

# Nitrate isotope investigations reveal future impacts of climate change on nitrogen inputs and cycling in Arctic fjords: Kongsfjorden and Rijpfjorden (Svalbard)

Marta Santos-Garcia[1], Raja S. Ganeshram[1], Robyn E. Tuerena[1, 2], Margot C.F. Debyser[1], Katrine Husum[3],
Philipp Assmy[3] and Haakon Hop[3]

[1]School of Geosciences, University of Edinburgh, Edinburgh, EH9 3FE, United Kingdom
[2]Scottish Association for Marine Science, Dunstaffnage, PA37 1QA, United Kingdom
[3]Norwegian Polar Institute, Fram Centre, Tromsø, 9296, Norway

*Correspondence to*: Marta Santos-Garcia (M.SantosGarcia@ed.ac.uk)

**Abstract.** Ongoing climate change in the Arctic has caused tidewater glacier retreat and increased discharge of freshwater and terrestrial material into fjords. These land inputs bring nutrients into the fjords and their cycling within the fjord system can vary with the influence of tidewater glaciers and the presence of sub-glacial meltwater plumes. In this study, we assess the influence of tidewater glaciers on nitrogen inputs and cycling in two fjords in Svalbard during the summer using stable isotopic

analyses of dissolved nitrate ($\delta^{15}N$ and $\delta^{18}O$) in combination with nutrients and hydrographic data. Kongsfjorden receives inputs from tidewater glaciers, whereas Rijpfjorden mainly receives surface inputs from land-terminating glaciers. Results showed that both fjords are enriched in nutrients from terrestrial inputs, where the inputs exceed Redfield ratios with excess Si and P relative to N. In both fjords, terrestrial nitrate from snowpack and glacier melting are identified as the dominant sources based on high $\delta^{18}O-NO_3^-$ and low $\delta^{15}N-NO_3^-$ of dissolved nitrate. In Kongsfjorden, mixed-layer nitrate is completely

consumed within the fjord system which we attribute to vigorous circulation at the glacial front influenced by the subglacial plume and longer residence time in the fjord. This is in contrast with Rijpfjorden where nutrients are only partially consumed perhaps due to surface river discharge and light limitation. In Kongsfjorden, we estimate terrestrial and marine N contributions to the nitrate pool from nitrogen isotopic values ($\delta^{15}N-NO_3^-$) and this suggests that nearly half the nitrate in the subglacial plume (50 ± 3 %) and the water column (44 ± 3 %) originates from terrestrial sources. In addition, we show that terrestrial N

also contributes significantly to regenerated N pool (63-88 %) within this fjord suggesting its importance in sustaining productivity within Kongsfjorden. Given this importance of terrestrial nutrient sources within the fjords, increase in these inputs due to climate change can enhance the fjord nutrient inventory, productivity and nutrient export offshore. Specifically, increasing Atlantification and warmer Atlantic Water will encourage tidewater glacier retreat and in turn increase surface discharge. In fjords akin to Rijpfjorden this is expected to foster more light limitation and less dynamic circulation, ultimately

aiding the export of nutrients offshore contributing to coastal productivity. Climate change scenario postulated for fjords such as Kongsfjorden include more terrestrial N-fuelled productivity and N cycling within the fjord, less vigorous circulation and the expansion of oxygen depleted deep waters inside the sill.

**Short Summary.** Terrestrial sources of nitrate are important contributors to the nutrient pool in the fjords of Kongsfjorden

and Rijpfjorden in Svalbard during the summer and they sustain most of the fjord primary productivity. Ongoing tidewater glacier retreat is postulated to favour light limitation and less dynamic circulation in fjords. This is suggested to encourage the export of nutrients to the middle and outer part of the fjord system, which may enhance primary production within and in offshore areas.



## 1 Introduction

In Arctic marine ecosystems, temperature anomalies of +2°C have led to increased discharge of freshwater (Beszczynska-Möller et al., 2012) along with fluxes of carbon and nutrients across the land-ocean interface with a range of implications for coastal ecosystems and biogeochemical cycling (McGovern et al., 2020). Nitrate is the predominant form of fixed N used by organisms in the ocean and is normally deemed as the limiting-nutrient in the Arctic Ocean (Yamamoto-Kawai et al., 2006), whereas phosphate and silica are also essential for algal growth in nearshore environments (Egge, 1998). Among these

nutrients, silicate and phosphate are mainly sourced from riverine inputs and are removed by biological uptake and sedimentation. Nitrogen has an additional source through fixation of atmospheric $N_2$ by microorganisms, while its sinks are mediated through denitrification and anaerobic ammonium-oxidising (anammox) processes (Sigman et al., 2001; Bristow et al., 2017). However, in fjords and coastal settings in the Arctic, terrestrial inputs of N could be important (Kumar et al., 2018) and these inputs are changing with polar warming through a myriad of factors such as increased river discharge, permafrost

thaw, soil N cycling and vegetation changes (Holmes et al., 2012). In addition, the nature of circulation and N-cycling can vary with the presence or absence of tidewater glaciers which are experiencing wide-spread retreat in the Arctic (Kohler et al., 2007; Østby et al., 2017). The purpose of the study was to document and identify the relative importance of terrestrial N sources and processes of N cycling in two fjords (Kongsfjorden and Rijpfjorden) in Svalbard through determinations of $\delta^{15}N$-$NO_3^-$ and $\delta^{18}O$-$NO_3^-$. This was used to elucidate the potential impacts of climate change on N cycling and exchange across the

land-ocean interface.

### 1.1 Using stable isotope tools to determine N fluxes and cycling

Biologically-mediated N transformation processes result in kinetic stable isotope fractionation often with unique fractionation factors which can be exploited to delineate these processes (Sigman and Fripiat, 2019). For instance, photosynthetic uptake

preferentially incorporates the lighter isotope $^{14}N$ as opposed to the heavier $^{15}N$, leading to kinetic fractionation (Ryabenko, 2013). The $^{15}N/^{14}N$ ratio resulting from that fractionation is presented as $\delta^{15}N$ (**Eqn.1**).

$\delta^{15}N$ (‰) = ((¹⁵N/¹⁴N) sample / (¹⁵N/¹⁴N) standard − 1) × 1000      **Eqn.1**
where ($^{15}N/^{14}N$)$_{standard}$ is the reference standard, measured in atmospheric $N_2$.


Isotopic fractionation caused by a given biological process is known as the kinetic effect, ε. According to **Eqn.2**, ε is defined by the difference in rates with which the two N isotopes are converted into product in such a manner that each N cycle process also has a characteristic isotope effect (Sigman and Casciotti, 2001).

ε (‰) = (¹⁴k / ¹⁵k − 1) × 1000        **Eqn.2**
where $^{14}k$ and $^{15}k$ are the rate coefficients of the reaction for $^{14}N$- and $^{15}N$ -containing reactant, respectively.

This ε value can be used to distinguish two types of fractionation during assimilation. These are, firstly, Rayleigh fractionation corresponding to a closed system, and secondly, steady state fractionation, which occurs in open systems where there is

resupply of nutrients (Sigman and Casciotti, 2001).

In addition, $\delta^{18}O$ values of N derivatives offer additional important constraints on natural processes (Kendall, 1998), where the $^{18}O/^{16}O$ ratio of nitrate is measured relative to the standard Vienna Standard Mean Ocean Water (VSMOW).



Dual isotope signatures of $\delta^{15}$N and $\delta^{18}$O can provide insight of nitrate inputs, such as the percentage contribution of terrestrial *vs.* marine sources (Sigman and Casciotti, 2001). The balance between $N_2$ fixation and denitrification renders mean $\delta^{15}$N and $\delta^{18}$O values of 5‰ and 2‰ respectively in the deep ocean, where $\delta^{15}$N ranges between ~1 and 20‰ (Brandes and Devol, 2002; Sigman et al., 2009a; Tuerena et al., 2015). Values higher than 5‰ ($\delta^{15}$N) and 2‰ ($\delta^{18}$O) in near surface waters results from fractionation during assimilation by phytoplankton at the ocean surface which enriches the residual pools of both $\delta^{15}$N-

$NO_3^-$ and $\delta^{18}$O-$NO_3^-$ (**Fig. A1a**; Altabet and Francois, 2001; Sigman and Casciotti, 2001). In comparison with marine nitrate, terrestrial sources of nitrate often have low values of $\delta^{15}$N-$NO_3^-$ and high values of $\delta^{18}$O-$NO_3^-$ subject to variable N inputs from N-fixation and atmospheric deposition on land (Heaton et al., 2004) and are further modified by soil processes such as denitrification and ammonia volatilisation to varying degrees (**Fig. A1a**; Carpenter et al., 1997; Kendall, 1998).

Isotopic analyses of $\delta^{15}$N and $\delta^{18}$O of nitrate can also provide quantitative estimates of the regenerated nitrate pool - as opposed to preformed nitrate (Sigman and Fripiat, 2019). The principle behind this is that regeneration moves the $\delta^{18}$O-$NO_3^-$ signature towards that of $\delta^{18}$O ($H_2O$) while conserving the $\delta^{15}$N-$NO_3^-$ signature (Granger et al., 2018) (**Fig. A1b**). Variations in $\delta^{15}$N-$NO_3^-$ : $\delta^{18}$O-$NO_3^-$ provide information about the $\delta^{15}$N-$NO_3^-$ from remineralisation (regenerated nitrate pool) and thus, also of the N exported and recycled from the euphotic zone (Sigman et al., 2005; Casciotti and Buchwald, 2012; Smart et al., 2015).

Processes that remove nitrate such as assimilation and denitrification fractionate ($\delta^{15}$N-$NO_3^-$ : $\delta^{18}$O-$NO_3^-$) in 1:1 ratio (**Fig. A1c;** DiFiore et al., 2009; Sigman et al., 2009b). The isotopic effects are best expressed in the residual nitrate pool when the processes are incomplete leaving the residual pool enriched as uptake preferentially utilises the lighter isotopes (Sigman and Casciotti, 2001). Mass balance dictates, on completion of these processes with complete utilisation of nitrate, the products of the reaction (biomass and $N_2$ gas respectively) reflecting the isotopic signatures of the initial nitrate source. This allows for the

assessment of relative nitrate utilisation by various processes.

### 1.2 Fjordic circulation

    The assessment of the impact of N fluxes, cycling and fate in a fjordic setting due to climate change is complicated by seasonally varying circulation patterns. Fjordic circulation responds to winter cooling and sea ice formation as well as summer

freshwater fluxes and in turn these processes also influence the nitrogen influxes and cycling and ultimate fate. Circulation in fjords is restricted owing to its narrow shape and presence of a sill (Svendsen et al., 2002; Dürr et al., 2011). The ineffective tidal mixing, alongside freshwater influxes, contributes to the development of a very sharp halocline and stratified water column during the summer months (Geyer and Ralston, 2011; Monteban et al., 2020). Deep convective mixing occurs in the autumn due to cooling of surface water (thermal convection) and in the winter due to sea ice formation and brine release (haline

convection) (Cottier et al., 2007). Remnants of winter cooled waters can persist into the summer and are characterised by low temperatures and a wide salinity range reflecting their variable origin in the outer basin or close to glacial fronts (De Rovere et al., 2022). In spring, temperature rises and ice break-up begins, leading to a reduction in salinity and the re-establishment of the strong summer pycnocline (Cottier et al., 2010).

Additionally, tidewater glaciers have a significant impact on fjord circulation as they contribute to additional freshwater flux from glacial melt (Cowan, 1992; Ingvaldsen et al., 2001). This subglacial discharge enters the fjord at depth- specifically at the glacial grounding line during the summer (Cowton et al., 2015; Carroll et al., 2015, 2016). As a result of its lower density, the submarine glacial melt forms an upwelling plume enhancing vertical mixing by the entrainment of fjord waters (Everett et al., 2018; Halbach et al., 2019). The plumes are strongest at the glacier fronts, and decrease with increasing distance from land

due to dilution and mixing (Darlington, 2015; Hopwood et al., 2020). Moreover, upwelling plumes can entrain and elevate





remnant winter cooled waters from the bottom of fjords, leading to the presence of a distinct, cold and relatively saline water mass throughout the water column close to tidewater glacier front (Torsvik et al., 2019). Recent studies have shown that meltwater plumes not only play a prominent role in vertical mixing at glacier fronts, but also enhance sub-surface lateral mixing (Torsvik et al., 2019). Importantly, plumes also promote the transfer of heat from the ocean to the glacier front of large

Greenland fjords, thereby drawing in warmer waters from offshore (Straneo et al., 2010; Cowton et al., 2015). However, in the smaller fjord systems in Svalbard, this invigorated lateral circulation is mainly confined within the fjords (Torsvik et al., 2019).

### 1.3 Study area

The sub-Arctic and Arctic fjords of Kongsfjorden (79.0°N, 11.7°E) and Rijpfjorden (80.0°N, 22.3°E), respectively, in Svalbard are ideal locations to document changes in N dynamics affected by freshwater influxes and the retreat of local glaciers. Important hydrographic features around Svalbard include the Fram Strait and the Barents Sea (**Fig. 1**). The Fram Strait is one of the main gateways into and out of the Arctic Ocean (Ilicak et al., 2016). On the western side of Svalbard, warm and salty Atlantic water (AW) is carried northward along the shelf edge by the West Spitsbergen Current (WSC) and enters the Arctic

Ocean, supplying Kongsfjorden and other fjords along the way (Cottier et al., 2005; Cokelet et al., 2008; Renner et al., 2018; Skogseth et al., 2020). These warm waters fill intermediate depths across the whole Arctic Basin and represent an important supply pathway for nutrients (Beszczynska-Möller et al., 2012; Randelhoff et al., 2018). Occasionally, AW protrudes onto the shelf area, and thus there is an interplay between the intrusion of AW at depth and meltwater at the surface (Onarheim et al., 2014).


Similarly, in the northern coast of Svalbard, the Atlantic influence remains substantial as AW travels eastward towards the Nansen Basin (Cokelet et al., 2008; Renner et al., 2018). However, fjords on the northern coast- including Rijpfjorden- have a wider continental shelf and are under a weaker AW influence compared to those on the western coast which have a narrower continental shelf (Hop et al., 2019). For instance, the mouth of Rijpfjorden is 60 km away from the shelf, whereas that of

Kongsfjorden is 45 km from the shelf (Howe et al., 2010). Rijpfjorden does experience occasional inflow of Atlantic-origin water during summer to late autumn (Wallace et al., 2010; Hop et al., 2019).

Kongsfjorden and Rijpfjorden not only experience different degrees of AW influence, but also have other contrasting physical characteristics which are relevant to N cycling. These characteristics include fjord size and geometry as well as the presence

of tidewater glaciers, all of which in turn affect circulation and residence times. Kongsfjorden is significantly deeper (350 m) than Rijpfjorden (c.a. 200 m), although their mean depths are similar (100 m) and so are their sizes (Kongsfjorden is 26 km long, 6-14 km wide; Rijpfjorden is 40 km long, 7-12 km wide). It is expected that the average water residence time in Rijpfjorden is less than 172 hours, the current estimate for Kongsfjorden (Torsvik et al., 2019; Yang et al., 2022). While Kongsfjorden experiences contributions of subglacial freshwater from five tidewater glaciers including Kongsbreen and

Kronebreen (**Fig. A2**; How et al., 2017) and direct runoff from the Bayelva river, Rijpfjorden only receives surface freshwater input from relatively small glacially-fed rivers with short-lived summer flow as inferred from satellite imagery (Wang et al., 2013). One important element of Kongsfjorden circulation is the subsurface discharge of meltwater from subglacial plumes (Darlington, 2015). The meltwater plumes play a prominent role in vertical and lateral exchanges, and consequently act as a nutrient pump - as documented by numerous studies (Darlington, 2015; How et al., 2017; Schild et al., 2017; Halbach et al.,

2019; Torsvik et al., 2019). Notably, the magnitude of the effect of subglacial discharge and the glacier front plume in Kongsfjorden varies spatially along the glacial front owing to topographical differences along the Kongsbreen and Kronebreen glacier fronts. In particular, the Kongsbreen front is deeper and narrower than that at Kronebreen, and this constrains the lateral



movement of water, thus the plume has a clearer vertical structure (Torsvik et al., 2019). In contrast, the Kronebreen front shows a less pronounced effect of the subglacial discharge and glacier front plume as lateral movement of water is unconstrained allowing mixing with adjacent fjord waters. Nonetheless, a topographic barrier below 35 m restricts lateral mixing below this depth along the Kronebreen transect (Torsvik et al., 2019). The differences in inflow regimes, the size, shape and the marine influence contribute to stronger light and nutrient limitation in Rijpfjorden as opposed to in Kongsfjorden (Hopwood et al., 2020).

Terrestrial N inputs exhibit high seasonality, often peaking in the summer months (McGovern et al., 2020), and have $\delta^{15}$N-$NO_3^-$ values typically ranging from 0 to 5‰ (Holmes et al., 2012; Sigman and Fripiat, 2019). Snowpack melting also has significant implications for the isotopic signatures. The snowpack acts as a sink for $^{15}$N-depleted atmospheric reactive nitrogen ($\delta^{15}$N-$NO_3^-$: ∼-6.5‰) (Heaton et al., 2004; Björkman et al., 2014; Vega et al., 2015). Other terrestrial inputs, such as tidewater glacier melting may contain N from bird faeces ($\delta^{15}$N-$NO_3^-$, ∼8‰; Skrzypek at al., 2015) and N from microbial degradation of organic matter in the glacier ice (D'Angelo et al., 2018), which can contribute to moderately N-rich meltwater runoff (Shi et al., 2018). Although $N_2$ fixation in diazotrophic bacteria has been reported in the Arctic regions (Blais et al., 2012; Zehr and Capone, 2020), the magnitude of this source remains poorly constrained. Low isotope values of $\delta^{15}$N-$NO_3^-$ have been attributed to terrestrial $N_2$ fixation in Svalbard (∼-0.5‰) (Skrzypek at al., 2015).

Ongoing "Atlantification" has caused the increased prevalence of warm AW displacing cold Arctic coastal waters off Svalbard (Polyakov et al., 2017), increasing sea water temperatures, shortening the sea ice-covered period, altering freshwater fluxes and enhancing light penetration (David and Krishnan, 2017; Hop et al., 2019; Hop and Wiencke., 2019). Significantly, freshwater discharge is expected to increase by 200% before 2100 in Svalbard (RCP4.5 scenario, Adakudlu et al., 2019), and as a result enhanced N inputs have already been reported in Svalbard and at a pan-Arctic scale (Holmes et al., 2012; McGovern et al., 2020).

Enhanced stratification due to freshwater inputs also impacts both terrestrial N supply and N cycling processes. It is therefore hypothesised that increased freshwater discharge from glacier retreat, snow and permafrost melt, retreating sea-ice cover and increasing riverine inputs will likely alter N supply and cycling in Kongsfjorden and Rijpfjorden. Alterations to both N inputs and cycling can thus lead to changes in the nutrient status and availability within the fjord, as well as altered exchange of nutrients between the fjord and offshore.

This study in Kongsfjorden and Rijpfjorden is aimed at understanding climate change impacts in N cycling; (i) allowing improved predictions of changes in the inventory of nutrients within sub-Arctic and Arctic fjords and, (ii) allowing an assessment of the future changes in exchange of nutrients between the fjords and offshore areas.

## 2. Methodology

### 2.1 Sampling sites

The sub-Arctic fjord of Kongsfjorden (79.0°N, 11.7°E; **Fig. 2**) and Arctic Rijpfjorden (80.0°N, 22.3°E; **Fig. 3**) are located on the islands of Spitsbergen and Nordaustlandet, Svalbard, respectively (**Fig. 1**). Kongsfjorden samples were collected during the Norwegian Polar Institute Monitoring Cruises on RV *Lance* between 29 July and 2 August 2017 (NP2017) and from 13 and 15 July 2018 (NP2018). These samples were from four areas, namely near glacier fronts (Kb5-7, only sampled during NP2018 cruise), fjord (Kb0-3), continental shelf (V12) and continental slope (V10 and V6) (**Fig. 2**).



Rijpfjorden was sampled between 3 and 5 August 2017 (NP2017) (**Fig. 3**), where sample locations are divided into inner fjord (R1,2), outer fjord (R3), continental shelf (R4-5) and continental slope (R6-7B) stations. The inner fjord and the outer fjord basins are separated by a shallow sill at around 40 m depth.

## 2.2 Nutrient and isotopic analysis

Water samples were collected from Niskin bottles mounted on a rosette equipped with a CTD system recording conductivity, temperature and pressure. In addition, other parameters measured were salinity (PSU), chlorophyll *a* fluorescence (mg m$^{-3}$) and oxygen (mL L$^{-1}$). Dissolved inorganic nutrient concentrations (nitrate, nitrite, silicate and phosphate) were analysed by the Norwegian Polar Institute (NP2017, NP2018) at the Institute of Marine Research. The samples were collected in 20 mL scintillation vials, fixed with 0.2 mL chloroform and stored refrigerated until sample analysis. Nitrite, nitrate, phosphate and

silicate were measured spectrophotometrically at 540, 540, 810, and 810 nm, respectively, on a modified Scalar autoanalyzer. The measurement uncertainty for nitrite was 0.06 mmol L$^{-1}$ and 10% or less for nitrate, phosphate and silicate. Water samples for isotopic analysis were prefiltered and frozen immediately after collection. The denitrifier method for dual nitrogen and oxygen isotopes was used for the isotopic analysis of dissolved nitrate (Sigman et al., 2001, McIlvin and Casciotti, 2011).

Nutrient and isotopic data from JR17005 and FS2018 cruise were also used in this study and were obtained from Tuerena et al. (2021a) and Debyser et al. (2022) respectively. This analysis was combined with nutrients and hydrographical data obtained from a conductivity-temperature-depth rosette and processed using Matlab_R2020a software to document N dynamics.

The denitrifier method takes advantage of the denitrifying bacteria *Pseudomonas aureofaciens* with limited N-reductase

activity which transform NO$_3^-$ to N$_2$O (Sigman et al., 2001; Casciotti et al., 2002; **Eqn. 4**). Nitrous oxide was extracted from 20mL-vial headspace by a Combi PAL auto-sampler and transported by a continuous flow of helium gas through a GasBench II coupled with a Delta V Advantage.

Two standards, USGS 34 and IAEA N3, were used as reference for isotope ratio mass spectrometry (IRMS) analysis. $\delta^{15}N_{AIR}$ values of these standards were 1.8 $\pm$0.2 (USGS 34) and 4.7$\pm$0.2 (IAEA N3), and $\delta^{18}O_{VSMOW}$ values were -27.9 $\pm$0.6 (USGS

34) and 25.6$\pm$0.6 (IAEA N3). Each standard was individually prepared at 2, 5, 15 and 30 $\mu$mol concentrations. To overcome the discrepancies in $\delta^{18}O$, solutions were prepared in milli-Q H$_2$O and low nutrient seawater respectively (McIlvin and Casciotti, 2011). Also, internal standards, from North Atlantic Deep Water, were used in order to represent average Atlantic Water $\delta^{15}N$ signature. This standard was run with each batch to check for inter-run comparability. IRMS analysis was carried out at the University of Edinburgh using ISODAT 2.5 software. Isotopic measurements were determined relative to a reference

peak.

Measurements of $\delta^{15}N$-NO$_3^-$ and $\delta^{18}O$-NO$_3^-$ were corrected to AIR and VSMOW, respectively, with the use of the correction scheme in Weigand et al. (2016) and following Tuerena et al. (2021a, 2021b) and the reference standards.

Averaged data reproducibility (1$\sigma$) across all runs was 0.3 and 0.5 for $\delta^{15}N$ and $\delta^{18}O$ respectively, determined using internal standards with isotopic values of 4.8 $\pm$ 0.2 ‰ ($\delta^{15}N_{AIR}$) and 1.8 $\pm$ 0.6 ‰ ($\delta^{18}O_{VSMOW}$). These deviations can be approximated

to normal variability (0.2 and 0.6‰ for $\delta^{15}N$ and $\delta^{18}O$, respectively).

## 2.3 Data analysis, processing and visualisation

Cross-section figures of Kongsfjorden were designed using a global Topo15.1 bathymetry dataset with a spatial resolution of 4 km (Smith and Sandwell, 1997). Bathymetric cross-section figures of Rijpfjorden are based on data from the Norwegian Mapping Authority Hydrographic Service and IBCAO database version 4.1.






The degree of stratification ($\Delta\rho$ , kg m$^{-3}$) was calculated as the difference in potential density between 10 and 40 m depth. In addition, apparent oxygen utilisation (AOU, µmol kg$^{-1}$) was computed using the equation AOU= [O$_2$]$_{saturated}$ − [O$_2$]$_{seawater}$. Meanwhile, the oxygen saturation (%) was given by [O$_2$]$_{seawater}$/[O$_2$]$_{saturated}$ × 100.

The semi-conservative tracers N* and Si* were calculated from inorganic nutrient concentrations where N* = NO$_x$ – 16 × PO$_4$ + 2.9 (Gruber and Sarmiento, 1997) and Si* = SiOH$_4$ – NO$_x$ (Sarmiento et al., 2004) are indicative of deviations from Redfield stoichiometry and 1:1 proportions, respectively. Negative N* values reflect N deficit and excess phosphate while positive N* values reflect excess nitrate relative to phosphate.

Regression analyses were computed to ascertain the significance of observed linear trends ($p$-value ≤ 0.05). All statistical analyses were included in Appendix B (**Table B1**).

### 3 Results

#### 3.1 Water mass characterisation and dynamics

Three different water masses were identified in Kongsfjorden, Rijpfjorden and their corresponding continental shelves and
slopes based on the temperature ($\theta$ in ºC) and salinity (S in PSU), following Pérez-Hernández et al. (2017).
Water masses in the continental slopes at both Kongsfjorden and Rijpfjorden were dominated by Atlantic Waters (AW; $\theta > 1$, S > 34.9) and to a smaller extent by Arctic Intermediate Water (AIW; $-1 \leq \theta \leq 1$, S > 34.9) (**Figs. 4, 5, A3**). Atlantic water dominance extended onto the continental shelf of Kongsfjorden (with a trough called "Kongsfjordrenna"). This AW intrusion is shown by salinity and temperature profiles above 40 m-depth, with salinity 34.9 PSU and temperature 5.6ºC (**Fig. A4**). This
intrusion was also associated with a chlorophyll *a* maximum of 4-6 mg m$^{-3}$, in agreement with recent studies reporting AW intrusions in Kongsfjorden (6.5 mg m$^{-3}$ chlorophyll maximum, Payne and Roesler, 2019).
In contrast, AW intrusion was weaker in Rijpfjorden due to the greater width of the shelf over northern Svalbard compared to the west coast. Although intrusion of AW was indicated by the branch of warmer water extending between 25 and 100 m depth at the continental shelf (**Fig. 5a**), the temperature was not warm enough to be characteristic of AW ($\theta < 1$,
**Fig. 5a**), nor the fluorescence peak (**Fig. 5c**) values were as high as those seen in the AW intrusion in Kongsfjorden (Payne and Roesler, 2019). Instead, the shelf at Rijpfjorden was dominated by Polar Surface Waters (PSW) (**Fig. A3**), which are a mixture of AW, river runoff, precipitation and ice melt (Rudels, 1989).

Within the fjords, two variants of Polar Surface Water (PSW) were also recognised, namely inshore PSW (iPSW) and warm
PSW (PSWw) (**Fig. A3**). In detail, iPSW is a mixture of ice melt and AW (iPSW, $\theta \geq 2$, S ≤ 34.9) while PSWw ($0 \leq \theta \leq 2$, S ≤ 34.9) is a mixture between PSW and warm AW (Rudels et al., 2005; Cokelet et al., 2008). Kongsfjorden was dominated by iPSW, whereas Rijpfjorden was governed by PSW, and in both fjords the presence of PSWw was limited.

The degree of stratification ($\Delta\rho$) in the upper 40m of the water column increases landwards in both fjords (**Table A1**; **Figs.**
**4b, 5b**) reflecting freshwater discharge. Notably, stratification was stronger in Rijpfjorden, particularly in the most offshore stations, compared to Kongsfjorden (**Table A1**). Moreover, while in Kongsfjorden freshwater was found in a thicker layer in the proximity of the tidewater glacier fronts (**Fig. 4b**), in Rijpfjorden, freshwater was confined to a fresher, thin surface layer that extended further offshore (**Fig. 5b; Fig. A2**). It is also worth noting that in Kongsfjorden, AW seemed to be drawn in at the continental shelf over the same depth range as the freshwater layer, which is a feature associated with lateral circulation in
fjords with tidewater glacier-induced plume set-up (Svensen et al., 2002; Cottier et al., 2005; Straneo et al., 2010; Cowton et al., 2015; Tverberg et al., 2019).



Additionally, cross-sections identified a fourth water mass, Winter Cooled Water (WCW, Cottier et al., 2005), in the deeper part of the fjord basins (c.a. >300 m at Kongsfjorden and c.a. >100 m at Rijpfjorden) characterised by low potential temperatures ($\theta$ <1.1ºC at Kongsfjorden, **Fig. 4a;** $\theta$ < -1ºC at Rijpfjorden, **Fig. 5a**), high apparent oxygen utilisation (AOU) of > 40 µmol kg$^{-1}$ (**Figs. 4d, 5d**) and oxygen saturation of < 90% (**Figs. 4e, 5e**). These features are indicative of water masses with the longest residence time and isolated from the atmosphere in both Kongsfjorden and Rijpfjorden (Svendsen et al., 2002) and are therefore sensitive to hypoxia with small changes in productivity, nutrient cycling and isolation time. Isolation and retention of such winter waters is attributed to reduced vertical exchange in the deep fjord basins, mainly due to restricted circulation owing to the presence of a sill at 200 m outside Kongsfjorden (in "Kongsfjordrenna") (**Fig. 4**) and at 25 m depth in Rijpfjorden (Cottier et al., 2010; **Fig. 5**).

## 3.2 Nutrient concentrations and isotopic ratios

Depth profiles of temperature, salinity, chlorophyll *a* fluorescence, nutrient concentrations, N*, Si* and isotopic ratios in Kongsfjorden and Rijpfjorden are illustrated in **Fig. 6a-j** and **Fig. 7a-j,** respectively. Moreover, cross-sections of nutrients, the semi-conservative tracers N* and Si* and isotopes ($\delta^{15}$N-NO$_3^-$ and $\delta^{18}$O-NO$_3^-$) are also shown in **Fig. 8a-g** and **Fig. 9a-g** for Kongsfjorden and Rijpfjorden respectively. In Kongsfjorden, temperature throughout the water column at the glacier front (Kb5-7, at Kronebreen and Kongsbreen transects) was strikingly lower than in the fjord and further offshore (**Fig. 6a**), indicating a distinct water mass formed at the front that cannot be traced offshore. It is suggested that these cold and saline waters are remnants of winter cooled waters resulting from heat loss to the atmosphere and contact with the glacier front (Torsvik et al., 2019; De Rovere et al., 2022). Only at Kronebreen front (**Fig. A2**, station Kb5), temperatures increased to values similar to fjord temperatures at 20-35 m depth (**Fig. 7a**). Here lateral movement is less constrained than in Kongsbreen front and, thus, mixing is possible (Torsvik et al., 2019). In contrast, in Rijpfjorden, temperature profiles did not show a distinct water mass at the fjord end (**Fig. 7a**). In addition, the halocline in Kongsfjorden was spread over a larger depth (~50 m-depth) and was less sharp than in Rijpfjorden (**Figs. 6b, 7b**), as supported by salinity cross sections (**Figs. 4b, 5b**). This is a characteristic feature of fjords with tidewater glaciers where plume set-up disperses salinity due to mixing and entrainment (Torsvik et al., 2019). In summary, Kongsfjorden exhibits the features of dynamic circulation associated with the subglacial plume set-up, which is absent in Rijpfjorden.

Nutrient and N* concentrations in Kongsfjorden were low or below the limit of detection (<1 µmol L$^{-1}$ nitrate, <0.2 µmol L$^{-1}$ phosphate, < -12.7 µmol L$^{-1}$ N* ) throughout the top 50 m in mid- to outer- fjord (Kb0-3) and continental shelf (V12) stations (**Figs. 6d,f,g, 8a,c,d**) due to uptake by phytoplankton as demonstrated by elevated chlorophyll concentrations (**Fig. 6c**). Such low nitrate and phosphate concentrations at these fjord stations were located within and above the halocline, whereas those at the continental shelf were related to the warmer AW intrusion (**Fig. 6a, b**). In Rijpfjorden, although lowest nutrient and N* concentrations were found above the halocline (~50 m-depth), in general they were higher with 1 µmol L$^{-1}$ nitrate, >0.2 µmol L$^{-1}$ phosphate and > -12.5 µmol L$^{-1}$ N* both in the fjord and at the shelf stations (**Figs. 7d,f,g, 9a,c,d**). These slightly higher nutrient concentrations in the stratified layers of Rijpfjorden may indicate incomplete nutrient utilisation at the surface as terrestrial inputs are unlikely to be much higher compared to Kongsfjorden.

In both fjords, coinciding with the base of the halocline, i.e., below 50 m depth, nitrate, silicate, phosphate and N* concentrations increased in all stations (**Figs. 6d-g, 7d-g**) while the isotopic ratios and Si* decreased (**Figs. 6h-j, 7h-j**). Decrease in nutrient concentration towards the surface accompanied by increases in $\delta^{15}$N-NO$_3^-$ and $\delta^{18}$O-NO$_3^-$ (**Figs. 6i,j, 7i,j**) are trends associated with uptake by phytoplankton, while the increase in N* with depth are associated with nutrient



regeneration. Higher Si* (> 0) towards the surface implies the presence of nutrient inputs carrying excess Si with respect to N

(**Figs. 6h, 8e**). In contrast, nutrient concentrations were higher in the glacier front at Kongsfjorden (Kb5-7) than in other fjord sampling sites. These high nutrient values were associated with low salinity (**Fig. 6b**), showed no defined depth-dependent trend (**Fig. 6d,f**) and Si concentrations were generally high enough to overcome Si-limitation while N-deficiency persisted (Si*>0 and N*<0; **Figs. 6g,h, 8d,e**). Moreover, $\delta^{15}$N-NO$_3^-$ was low and clustered at around 4.3 $\pm$ 0.1‰ (**Fig. 6i**) indicating a source dependence rather than uptake.

Conversely, nutrient and N* concentrations reached 3.2μmol L$^{-1}$ nitrate, 0.3μmol L$^{-1}$ phosphate and -10.4 μmol L$^{-1}$ N* at 25m-depth in the continental slope at Kongsfjorden (**Figs. 6d,f,g, 8a,c,d**), decreasing towards the surface. There, elevated $\delta^{15}$N-NO$_3^-$ and $\delta^{18}$O-NO$_3^-$ with increasing proximity to water surface (**Figs. 6i,j, 8f,g**) are explained by uptake by phytoplankton. The same interpretations also apply to the continental slope of Rijpfjorden (**Figs. 7d-j, 9a-g**).

        In Rijpfjorden, $\delta^{15}$N-NO$_3^-$ and $\delta^{18}$O-NO$_3^-$ values taken at the deepest sampling depth of 100 m showed enrichments

in isotopic values from the shelf and slope stations into the fjord. Namely the $\delta^{15}$N-NO$_3^-$ at fjord stations was 5.8 $\pm$ 0.2 ‰, which was higher than the shelf and slope signature of 5.4 $\pm$ 0.4 ‰ (**Fig. 7i**). Meanwhile, $\delta^{18}$O-NO$_3^-$ at fjord stations (2.7 $\pm$ 0.5 ‰) also showed an enrichment with respect to shelf and slope stations (2.2 $\pm$ 0.7 ‰) (**Fig. 7j**). Noticeably, $\delta^{15}$N-NO$_3^-$ and $\delta^{18}$O-NO$_3^-$ signatures at shelf and slope stations fall within the range of AW signatures ($\delta^{18}$O$_{AW}$-NO$_3^-$ = 2.8 $\pm$ 0.3 ‰; $\delta^{15}$N$_{AW}$-NO$_3^-$ = 5.1 $\pm$ 0.1 ‰; Tuerena et al., 2021b).


## 4 Discussion

### 4.1 Evidence for terrestrial nutrient inputs

In Kongsfjorden, higher nutrient concentrations at the glacier front stations were associated with nutrient inputs from glacial

discharge plumes. Concentrations of 1-4 μmol L$^{-1}$ nitrate, 0.2-0.4 μmol L$^{-1}$ phosphate and 1.5-2.5 μmol L$^{-1}$ silicate found within the halocline in stations near glacier fronts can only be explained by this terrestrial supply as the halocline presence clearly restricts nutrient mixing from depths into the upper 50m -as was shown in nutrient profiles in other fjord stations (**Figs. 6d-e, 7d-e**). Indeed, glacial meltwater has been found to be enriched in these nutrients, with concentrations broadly ranging from 0.24- 1.60 μmol L$^{-1}$ nitrate, 0- 0.19 μmol L$^{-1}$ phosphate and 1.72- 3.47 μmol L$^{-1}$ silicate (Halbach et al., 2019). In Rijpfjorden,

where nutrient concentrations were generally higher than in Kongsfjorden, there are no tidewater glaciers, but this does not rule out the presence of terrestrial inputs from surface meltwater flow through streams. In fact, terrestrial inputs are potentially responsible for the enrichment in silicate (>1μmol L$^{-1}$) and phosphate (>0.2 μmol L$^{-1}$) in inner Rijpfjorden. Moreover, marine inputs cannot explain higher nutrient concentrations in Rijpfjorden as AW intrusion is too weak to extend to the continental shelf.


        On the continental shelf outside Kongsfjorden, a strong thermocline at ~40 m pointed to a strong marine influence through the intrusion of AW between 20-50 m (34.9 PSU, 5.6ºC) into the fjord. This intrusion was associated with a chlorophyll *a* maximum of 4-6 mg m$^{-3}$ and nutrient-depletion due to phytoplankton growth, thus suggesting a potential marine contribution to the nutrient pool of Kongsfjorden. Vertical supply of nutrients from underlying modified PSW (34.7 PSU, 3ºC) was likely

hindered by the strong thermocline. Likewise, outside of the sill, AW was well-mixed and dominated throughout the whole water column, hosting a phytoplankton bloom with a fluorescence peak (14 mg m$^{-3}$) at 20 m depth.

Marine nutrient contribution to productivity in both fjords is hindered by the strong halocline that develops in summer where mixing is restricted to winter overturning. In addition, AW is relatively nutrient poor as nutrient depletion occurred before





entering the fjord (Cottier et al., 2010). Therefore, terrestrial nutrient inputs which occur throughout the summer with higher
        nutrient concentrations than marine inputs can be more important in these fjords during the summer (Hopwood et al., 2020).

        Terrestrial inputs, evident from glacier front profiles, can be distinguished using nutrient stoichiometry by plotting nitrate
        concentrations against phosphate (**Fig. 10a**) and against silicate (**Fig. 10b**) concentrations in the whole of Kongsfjorden and
Rijpfjorden.
        N and P concentrations in Kongsfjorden, Rijpfjorden and eastern Fram Strait are shown in **Fig. 10a** along with lines
        representing Redfield ratio (N:P, 1:15 and 1:16). Linear trends with slope similar to Redfield ratios indicate phytoplankton
        uptake and regeneration stoichiometry. Thus, most of the fjord samples were influenced by nutrient uptake and recycling (**Fig.
        10a**; $\Delta N/\Delta P$ at Kongsfjorden and Rijpfjorden was 16.8; $R^2$ 0.96, *p*-value $\leq$ 0.05) as slope conforms broadly to Redfield ratio.
However, the x-intercept (0.14 $\mu$mol $PO_4^{3-}$ $L^{-1}$) suggests surplus supply of phosphate from riverine input into the fjords and
        the inner shelf area (**Fig. 10a;** McGovern et al., 2020). It has been reported that discharge from the Bayelva river into
        Kongsfjorden contributes 60 $\mu$mol $PO_4^{3-}$ $L^{-1}$, equivalent to 5.4 tons per year using a total discharge of 29 x $10^6$ $m^3$ recorded in
        2012 (Zhu et al., 2016), whereas Rijpfjorden receives P-input by glacially fed rivers (Wang et al., 2013). In general, nutrients
        supplied from Arctic soils are enriched in phosphate relative to nitrate due to the loss of N via denitrification owing to
waterlogging during the summer months (Hayashi et al., 2018).

            In contrast, nitrate and phosphate measured in eastern Fram Strait cannot explain the P intercept recorded in
        Kongsfjorden and Rijpfjorden (i.e., 0.14 $\mu$mol $PO_4^{3-}$ $L^{-1}$) as it has an intercept of $\sim$ 0 and is consistent with recent studies in
        the AW mixed layer of Fram Strait (Tuerena et al., 2021a; **Fig. 10a**). Thus, the nutrient stoichiometry of Kongsfjorden and
Rijpfjorden are influenced by terrestrial nutrient inputs. Broadly, despite this terrestrial input, the nutrient data plotted in **Fig.
        10a** indicates rapid N depletion in the upper water column of both fjords. This can be explained by stratification and excess
        terrestrial P inputs relative to the Redfield ratio.

        Inshore and fjord samples that show excess P also show higher Si:N ratios due to terrestrial Si supply both in Kongsfjorden
and Rijpfjorden (**Fig. 10a**; Dugdale and Wilkerson, 2001). When nitrate *vs.* silicate concentrations are plotted using salinity as
        a third variable, most stations follow a linear trend with a N:Si ratio of 2.5 (**Fig. 10b**). Outliers are associated with low salinity
        (<33.5 PSU) and have a lower N:Si ratio of 1.3 indicative of Si enrichment (relative to $NO_3^-$) from terrestrial discharge (**Fig.
        10b**). This is supported by Si* profiles, where values are higher within the halocline (**Figs. 6g, 8e**). These outliers correspond
        to innermost stations at both fjords and are represented as high Si:N ratios (> 1) in **Fig. 10a**. Furthermore, in Kongsfjorden,
data also suggest that terrestrial silicate is supplied through the glacial discharge plume (**Fig. 10b**), most likely from glacier
        meltwater enriched in silicate from weathering of siliceous rocks (Dugdale and Wilkerson, 2001; Halbach et al., 2019).
        Meanwhile, in Rijpfjorden, silicate is supplied through glacially fed rivers rather than directly by tidewater glaciers (Wang et
        al., 2013). In contrast, AW is a poor source of silica, with silicate limitation (Krisch et al., 2020) and phosphate deficiency
        evident in surface waters (Tuerena et al., 2021a) and make only limited contribution to the halocline in both fjords. Therefore
it is suggested that Kongsfjorden and Rijpfjorden have the potential to act as a source of silicate and to a lesser extent phosphate
        to AW. This aspect is discussed below taking into consideration the fjord residence times.

### 4.2 Terrestrial inputs and nutrient limitation

        Both Kongsfjorden and Rijpfjorden receive terrestrial nutrients enriched in Si and P relative to N, via glacial plumes and
riverine input, which are then consumed by phytoplankton uptake within the fjord (**Fig. 10**). Here we evaluate the potential
        impact of this on siliceous diatoms (Egge, 1998), by comparing Si:N uptake ratios within *vs.* outside of the fjord (**Fig. 11**).



In both fjords, data from surface waters (<100 m) indicate an Si:N uptake ratio of 0.3 ($R^2$= 0.74, $p \leq 0.05$), and in the eastern Fram Strait of 0.4 ($R^2$= 0.84, $p \leq 0.05$) (**Fig. 11**). This suggests that the diatom production as a proportion of the whole

phytoplankton community is higher in the eastern Fram Strait (c.a. 40% of phytoplankton are diatoms) compared to fjord stations (ca. 30%) given that diatom Si:N uptake ratio is ~ 1 (Brzezinski, 1985). This implies that terrestrial Si inputs have no significant impact on phytoplankton composition. This is probably because terrestrial Si inputs are insufficient to overcome the Si limitation as all fjordic samples fall below 5 $\mu$molSi L$^{-1}$, which is generally thought to be the threshold for kinetic Si limitation (Krause et al., 2018; **Fig. 11**). Si limitation in the fjord can be explained by the "silicate pump", whereby more Si is

rapidly lost to the deep- through sinking of biogenic silica or $Si(OH)_4$-rich faecal pellets- compared to N, and Si resupply is further prevented by stratification (Dugdale et al., 1995).

Several studies have supported higher diatom abundance outside of the fjord as diatoms generally dominate the phytoplankton assemblage in Arctic spring blooms (Hodal et al., 2012; Krause et al., 2019). Uptake ratio reported herein (ca. 0.4) in eastern

Fram Strait suggests phytoplankton succession and a shift from diatom dominance in the spring bloom to nano- and picoplankton communities in summer months (Strom et al., 2006). Such phytoplankton succession is triggered by the depletion of nutrients from surface waters following the spring bloom and has been previously reported outside of Kongsfjorden (Rokkan Iversen and Seuthe, 2011; Kulk et al., 2018). Previous studies have documented that small nano- and picophytoplankton dominate the phytoplankton community throughout the summer in Kongsfjorden (Piquet et al., 2014; Kulk et al., 2018),

although phytoplankton abundances can be patchy and succession can vary from year to year due to sea ice duration and hydrological conditions. There are reports of diatoms being more abundant in cold years and small flagellates during warmer years (Piwosz et al., 2015). The results presented here therefore reflect an integration over the summer growing season sampled. However, in general, small phytoplankton cells thrive in fresher and oligotrophic surface waters due to their large surface-area-to-volume ratio which provides effective acquisition of nutrient solutes and photons, and hydrodynamic resistance

to sinking (Li et al., 2009).

Thus, our results document terrestrial nutrient inputs within the fjords, P inputs occur in excess of the Redfield ratio (N:P=16) and the excess terrestrial Si inputs do not necessarily by itself favour greater diatom production due to nutrient limitation within the fjords.

**4.3 Sources of Nitrogen and their isotopic composition**

In a fjordic setting, various sources of terrestrial nitrogen are discharged and mixed with marine sources, subjected to varying degrees of uptake and recycling with the fjords and the residual exported to the shelf. Dual isotope signatures of $\delta^{15}N$-$NO_3^-$ and $\delta^{18}O$-$NO_3^-$ as well as nitrate concentrations are used to delineate the relative importance of these processes. $\delta^{15}N$-$NO_3^-$ and $\delta^{18}O$-$NO_3^-$ were used to trace nitrogen cycling processes as they exhibit characteristic isotopic fractionation trends (Sigman

and Fripiat, 2019) and offer insight into the relative importance of potential nitrate sources (Sigman and Casciotti, 2001).

In **Figure 12**, $\delta^{15}N$-$NO_3^-$ is plotted against the log of nitrate concentration with salinity as a third variable. If biological uptake-dominated changes in nitrate concentration and composition, data would show a linear relationship, as expected from Rayleigh fractionation, and the slope of the line would represent the fractionation factor ($\epsilon$; Altabet and Francois, 2001). Fractionation

during uptake would lead to $\epsilon$ within the range 3-10 (Wankel et al., 2009). Indeed, $\epsilon$ of 4.9‰ was recorded in the Fram Strait (Tuerena et al., 2021b) and for comparison the trend line from this Fram Srait study is shown as a dotted line in **Figure 12**. In Kongsfjorden and Rijpfjorden, most fjordic samples do not show the linear trend conforming to Rayleigh fractionation. Deviation from the linear relationship increases landwards within the fjord, where innermost stations show a slope near 0 (**Fig.**



**12**). This suggests that the isotopic signatures of nitrate are not governed by assimilation processes alone and that mixing

between marine and terrestrial nitrate sources influence isotopic signals significantly. The degree of stratification, and its associated nutrient dilution effect, can be ruled out as salinity alone could not explain the isotope effect in **Figure 12**. Additionally, the terrestrial source is significantly lower in $\delta^{15}$N as the trends indicate lighter signatures with proximity to land in both fjords (**Fig. 12**). Importantly, the lowest $\delta^{15}$N signatures with no slope are found in the Kongsfjorden glacial front samples where terrestrial inputs of P and Si are identified (**Fig. 10**). The remnants of winter cooled waters at Kronebreen and

Kongsbreen (Halbach et al., 2019; Hopwood et al., 2019) have distinctly lighter isotopic values ($\delta^{15}$N-NO$_3^-$ of 3.9-4.3 ‰) relative to the rest of fjordic samples indicating a larger contribution from terrestrial sources rather than marine sources (AW $\delta^{15}$N-NO$_3^-$ of 5.1 $\pm$ 0.1 ‰, Tuerena et al., 2021b) causing significant deviations from uptake dominated fractionation trends. Meanwhile, the epsilon value of remaining fjordic samples ($\varepsilon$= 1.8‰) can be approximated to that in salinity stratified PSW and in northern Barents Sea ($\varepsilon$= 2‰; Tuerena et al., 2021b).


A plot of $\delta^{15}$N-NO$_3^-$ *vs.* $\delta^{18}$O-NO$_3^-$ with depth as the third variable is shown in **Fig. 13a**. Data from Kongsfjorden and Rijpfjorden show three clear trends. Firstly, data from Kongsfjorden -with the exception of glacier front stations- and data from the continental slope outside of Rijpfjorden showed a positive correlation (R$^2$ 0.98, *p*-value ≤ 0.05) with a slope close to 1, whereby isotopic signatures became heavier at shallower depth (**Fig. 13a**). This is indicative of nitrate assimilation by

phytoplankton which fractionates both isotopes equally, that is $\delta^{15}$N-NO$_3^-$: $\delta^{18}$O-NO$_3^-$ ratio approaching 1 (DiFiore et al., 2009; Sigman et al., 2009b). Although it is inferred that biological uptake mainly determines the isotopic trends in these samples, the slight deviation from 1 (c.a. 0.8) may also result from simultaneous uptake and recycling. During recycling, $\delta^{18}$O of nitrate gets reset closer to water values with an added isotopic fractionation effect of 1.1‰ (Buchwald et al., 2012; Sigman et al., 2009b). In these fjordic settings, $\delta^{18}$O$_{H2O}$ is estimated to be c.a.-0.3‰ according to the equation $\delta^{18}$O$_{H2O}$= 0.43S-14.65 derived

for Kongsfjorden (MacLachlan et al., 2007) using the average salinity (S) of 34.8 $\pm$ 0.006 PSU in the winter cooled waters. For comparison purposes, another $\delta^{15}$N-NO$_3$ *vs.* $\delta^{18}$O-NO$_3$ plot is presented (**Fig. 13b**) using only stations from the continental shelf and slope of Kongsfjorden as well as from eastern Fram Strait. Data show a positive correlation (R$^2$ 0.77, *p*-value ≤ 0.05) with a slope of 0.8, with $\delta^{18}$O-NO$_3$ values clustering at 2.0 $\pm$ 0.6‰ and at 4.9 $\pm$ 0.2 ‰ for $\delta^{15}$N-NO$_3^-$; these values are indicative of the initial isotopic signatures of the marine endmember. A comparison of **Fig. 13a** and **Fig. 13b**, reveals that the slope of

the line from Kongsfjorden samples is similar to the slope of offshore and Fram Strait samples (~0.8). In summary, biological uptake is the most likely determinant of the isotopic trends in these samples.

However, the glacier front stations in Kongsfjorden deviated significantly from the 1:1 line, with $\delta^{15}$N-NO$_3^-$ values clustering around 4.3 $\pm$ 0.1‰, meanwhile, $\delta^{18}$O-NO$_3^-$ ranged from 2.3 to 6.5 ‰ (**Fig. 13a**). This deviation was caused by relatively light

$\delta^{15}$N-NO$_3^-$ and enriched $\delta^{18}$O-NO$_3^-$ signatures, which were derived from terrestrial sources, in comparison to other fjordic stations. In addition to the deviation from 1:1 relationship, isotopic values of these samples did not show any linear trend. This indicates source dominance rather than uptake. These samples were previously identified within the glacial discharge plume (S <33.5PSU) with high silicate values deviating from the Si:N line (**Fig. 10b**). These samples incorporate terrestrial nitrate source signatures which are rapidly lost away from the glacial plume through biological uptake.


In **Fig. 13a**, fjord and continental shelf samples at Rijpfjorden also deviated from the 1:1 line – falling on a slope of 0.6 (R$^2$ 0.81, *p*-value ≤ 0.05). Interestingly, the extension of the line of best fit of Rijpfjorden data crosses the cluster of data points comprising glacier front Kongsfjorden stations (i.e., the terrestrial endmember signature). However, the samples themselves do not fall within the cluster and instead show enriched isotopic signatures. This is indicative of the input of

terrestrial nitrate in Rijpfjorden with relatively light $\delta^{15}$N-NO$_3^-$ and enriched $\delta^{18}$O-NO$_3^-$ signatures similar to the glacial plume samples of Kongsfjorden but subsequently modified by uptake and recycling as described below.





Unlike Kongsfjorden, nitrate is not completely depleted in Rijpfjorden indicating partial nitrate utilisation which leads to heavy residual nitrate signatures where $\delta^{18}O\text{-}NO_3^-$ reach values as high as 14‰ and $\delta^{15}N\text{-}NO_3^-$ up to 10‰ (**Fig. 13a**). The lower

gradient (0.6 instead of 0.8) is indicative of partial uptake and regeneration. The partial uptake leads to heavy signatures of $\delta^{15}N\text{-}NO_3^-$ and $\delta^{18}O\text{-}NO_3^-$ but simultaneous recycling forces $\delta^{18}O$ towards lower values close to $\delta^{18}O_{H2O}$. This $\delta^{18}O_{H2O}$ can be assumed to be the same in both fjords (0.3‰). Thus, isotopic composition reveals the importance of the terrestrial nitrate source in both fjords as well as differences between the two fjords in the way nitrate is consumed and recycled.

In Kongsfjorden, nitrate uptake is complete with near-zero nitrate values above the halocline with the exception of those in proximity to the subglacial plume at the glacial front. When nitrate is present it reflects the additional terrestrial source and its isotopic signature. In Kongsfjorden, nitrate is present at the glacial front where the sub-glacial plume supplies terrestrial nutrients to the fjord; however this nitrate is rapidly consumed within the fjord with near-zero nitrate values above the halocline within the fjord. In contrast, nitrate is not fully consumed in near surface waters of Rijpfjorden leading to significant isotopic

fractionation. There, the isotopic trends in **Figure 13** -where the fractionation line passes through the glacial plume samples representing terrestrial endmember in this setting- suggest significant terrestrial nitrate contribution in Rijpfjorden which is subsequently masked by mixing with partially utilised nitrate. This partly explains why the terrestrial source signatures with low $\delta^{15}N\text{-}NO_3^-$ signature, evident in Kongsfjorden, are not as obvious in Rijpfjorden (**Fig. 13a**). The difference in nitrate utilisation suggested above in these two fjords can arise from the nature of circulation in these fjords which is discussed in

detail below in the context of the role of subglacial plume in nutrient dynamics.

**4.4 The role of the subglacial plume in nitrate use and cycling**

The difference in nitrate utilisation suggested above in these two fjords can arise from the nature of circulation in these fjords

and the difference in light limitation resulting from surface and subsurface discharge. In Rijpfjorden, the fjord stations show strong freshwater stratification, and the fresher water is confined to a shallow surface layer resulting from surface discharge from rivers (**Figs. 5b, 7b**). In contrast, at the glacial front of Kongsfjorden meltwater influences a larger depth in the water column with less fresh surface waters (**Figs. 4b, 6b, A3**). This reflects the presence of the subglacial plume in Kongsfjorden where the plume rises from the grounding line of tidewater glaciers at 60 m depth (Darlington, 2015) increasing mixing

between meltwater and the entrained saltwater over a larger depth range near the surface (Everett et al., 2018). This is agreement with a previous study which documented the plume to extend to 40 m depth in this fjord (Torsvik et al., 2019). It is worth noting that the meltwater plume in Kongsfjorden is mainly subglacial freshwater as opposed to glacial surface melt (How et al., 2017).

Importantly, the magnitude of the effect of the subglacial discharge and glacier front plume in Kongsfjorden varies spatially along the glacial front owing to topographical differences along Kongsbreen (Kb6-7) and Kronebreen (Kb5) transects. Along both fronts, a distinct cool and saline water mass was identified as remnant winter cooled waters formed due to heat loss to the atmosphere and contact with the glacier front (De Rovere et al., 2022). Studies by Torsvik et al. (2019) attributed the presence of this water mass in Kongsfjorden to its entrainment and elevation by the glacial plume dynamics. However, along Kronebreen

front these winter cooled waters undergo mixing with fjord water as shown by temperature above 3.5° C and salinity above 33.5 PSU at 20-35 m depth (**Fig. 6a, b**) and the presence of the subglacial discharge at glacier front plume is less distinct. At Kronebreen front, mixing associated with the plume is topographically unconstrained -as opposed to in Kongsbreen – as it is wider and thus allowing lateral water movement (Torsvik et al., 2019). The lack of mixing below 35m-depth is explained by a topographic barrier that restricts the exchange of water masses as suggested by particle analysis (Torsvik et al., 2019).




Whether the release of glacially eroded, entrained sediments (Elverhøi et al., 1983; Trusel et al., 2010; Kehrl et al., 2011) occurs through surface or subsurface discharge can impact light limitation. Rijpfjorden experienced direct surface discharge release through glacially fed rivers (Wang et al., 2013). This surface release of sediments is expected to encourage stronger light limitation at the surface than subsurface discharge, as suspended sediments remain near the surface due to strong

stratification (Cowton et al., 2015; Carroll et al., 2015, 2016). Near-zero fluorescence values in the inner fjord of Rijpfjorden support that productivity could be hindered by light limitation. In contrast, in Kongsfjorden, turbidity associated with sediment discharge is subject to dynamic mixing associated with subglacial plumes (Cowton et al., 2015; Carroll et al., 2015, 2016) which invigorate vertical mixing and lateral exchange of water. In such a dynamic setting, with weak stratification, the turbidity caused by glacially eroded sediments is relatively more dispersed, increasing light penetration away from the plumes which

facilitates plankton blooms (Calleja et al., 2017). Such tidewater glacier-related circulation dynamics have been documented in Kongsfjorden (Darlington, 2015; How et al, 2017; Schild et al., 2017; Halbach et al., 2019). The longer residence times and the dynamic circulation associated with tidewater glaciers in Kongsfjorden lead to complete utilisation of nutrients at the surface and recycling at depth within the fjord (i.e., increased regenerated nutrient storage), inhibiting the export of nutrients offshore. In Rijpfjorden, light limitation retards nutrient utilisation (Torsvik et al., 2019) leading to only partial use as suggested

by the isotope data. Moreover, the differences could also result from delayed nutrient utilisation in Rijpfjorden compared to Kongsfjorden due to its more extensive ice cover. Other additional factors could be the shallower depth of Rijpfjorden and thus shorter residence time which facilitates shelf exchange before nutrients are fully utilised (Straneo and Cenedese, 2015; Morlighem et al., 2017). This increases the opportunity for exporting the heavy, partially-utilised nitrate from Rijpfjorden into the shelf where it can be subsequently utilised and impact productivity.


As a final note, the scenario described for Rijpfjorden is representative of what is happening in Greenland and other Arctic fjords that are currently experiencing tidewater glacier retreat onto land (Nuth et al., 2013; Meire et al., 2017; Kanna et al., 2018). The implication is that, with increasing Atlantification and warmer AW in the future, AW intrusion in Arctic fjords will strengthen and thus encourage tidewater glacier retreat. This will favour surface discharge of terrestrial nutrients, less dynamic

circulation, leading to light limitation and retarded use of nutrients and ultimately, aiding the export of nutrients offshore. The significance of this stems from the fact that the Greenland ice sheet and its associated tidewater glaciers have been suggested to be an important source of nutrients to the wider Arctic Basin and thus exert an important control on overall Arctic net primary production (Hawkings et al., 2015), with terrestrial nutrient inputs currently estimated to support 28-51% of net primary production basinwide (Nuth et al., 2013; Terhaar et al., 2021). It is worth noting some differences between Svalbard

and Greenland settings. Some Greenland fjords have very deep tidewater glaciers and receive smaller terrestrial N inputs due to less soil, thus marine sources of nutrients could be relatively more important (Cape et al., 2019).

### 4.5 Isotopic fingerprinting of terrestrial and nitrogen sources

The isotopic studies indicate significant terrestrial N inputs in both fjord systems considered. In Kongsfjorden the glacier front

samples identified with terrestrial sources of nitrate have isotopic signatures that do not show evidence for uptake and isotopic fractionation. These samples have $\delta^{15}$N-NO$_3^-$ values clustering around 4.3± 0.1‰, meanwhile, $\delta^{18}$O-NO$_3^-$ ranges from 2.3 to 6.5 ‰. Although the isotopic signature of the terrestrial N is masked by mixing and partial use of nitrate in Rijpfjorden, the isotope fractionation trendline passes through the glacial front samples of Kongsfjorden, indicating similar terrestrial source signatures in both fjords (**Fig.13a**). This provides the opportunity (1) to identify the origin of terrestrial sources and (2) to

delineate terrestrial and marine component of N in the upwelling plume at the glacial front.



High $\delta^{18}$O-NO$_3^-$ values (60-86‰; Heaton et al., 2004) have been reported within atmospherically-deposited nitrate related to ozone-depleted air in Svalbard's snowpack (Björkman et al., 2014). Meanwhile low $\delta^{15}$N-NO$_3^-$ in snowfall has been atrributed to long range atmospheric transport deposition over the Arctic Ocean ($\delta^{15}$N$_{snowfall}$ of -4‰; Heaton et al., 2004; Vega et al.,
2015). Thus, nitrate from this source is expected to have lighter $\delta^{15}$N and heavier $\delta^{18}$O relative to marine values ($\delta^{18}$O-NO$_3^-$ of 2.0 $\pm$ 0.6‰ and $\delta^{15}$N-NO$_3^-$ of 4.9 $\pm$ 0.2 ‰; **Fig. 13b**). In this regard it is important to note that during warm summers as much as 50% of the annual snowpack accumulation in Svalbard may melt (Pohjola et al., 2002). Therefore it is expected that the dominant terrestrial source could be melting of seasonal snowpacks and glacier ice because it acts as a sink for atmospheric reactive nitrogen with high $\delta^{18}$O-NO$_3^-$ and low $\delta^{15}$N-NO$_3^-$ signatures as previously mentioned (Björkman et al., 2014; Vega et
al., 2015). The predominance of this source is consistent with isotopic composition (lighter $\delta^{15}$N and heavier $\delta^{18}$O) of nitrate in Kongsfjorden glacial front and from the intersection of the line of best fit at Rijpfjorden through this data (**Fig. 13a**).

Other potential terrestrial sources can also have a minor impact and these include glacier melting, river run-off, bird guano and moulins on glaciers (**Fig. A2;** Hop et al., 2002; Lydersen et al., 2014). Nitrate derived from bird guano is expected
to isotopically heavy, as nitrification of guano-derived NH$_4$ would have enriched the $\delta^{15}$N-NO$_3^-$ signature in the fjord (c.a. 8 ‰; Bokhorst et al., 2007; Szpak et al., 2012; Skrzypek et al., 2015). Additional terrestrial nitrate sources include microbial degradation of organic matter within the snowpack and particle-bound ammonium (Halbach et al., 2019) associated with the transport of fine sediments from subglacial erosion and Bayelva river (Hodson et al., 2005). These additional sources are likely to be relatively small, but can have an impact on the isotopic signature of terrestrial sources.


Overall, the 4.3$\pm$ 0.1 ‰-$\delta^{15}$N-NO$_3^-$ value (**Fig. 13a**) could be considered as nitrate sourced from the upwelling plume. The influence of biological uptake in the glacier front discharge plume is expected to be low as high suspended matter hinders light penetration and phytoplankton production in the plume regions (Kumar et al., 2018). This is supported by very low chlorophyll contents (<0.8 mg m$^{-3}$, this work), and high mineral matter load in GF/F filters supports this contention. The fact that the $\delta^{15}$N-
NO$_3^-$ values strongly cluster around 4.3 ‰ without any relationship to nitrate concentration as would be expected from uptake also support this (**Fig. 12**). However, the plume dynamics involve some degree of mixing with marine sources due to entrainment of underlying marine waters. This mixing is evaluated below.

Published terrestrial endmember values lie at 3.5 ‰ -$\delta^{15}$N-NO$_3^-$ from surface sediments at innermost Kongsfjorden (Kumar et al., 2018). This number is derived from dual C and N isotope mixing model based on Kongsfjorden sediments. Thus
the 4.3 ‰ value evident for $\delta^{15}$N-NO$_3^-$ at the fjord front can be regarded as an admixture of terrestrial N (c.a., 3.5 ‰; Kumar et al., 2018) and marine source (c.a., 5.1 $\pm$ 0.1 ‰; Tuerena et al., 2021b) in equal proportions ($\frac{4.3-3.5}{(4.3-3.5)+((5.1\pm0.1)-4.3)} \times 100 = 50 \pm 3$ %). Note that the marine endmember of 5.1 $\pm$ 0.1 ‰ (Tuerena et al., 2021b) is used in the mixing calculation instead of that estimated from continental shelf and slope samples (4.9 $\pm$ 0.2 ‰; **Fig. 13b**) as it is a better representation of pure AW signal, while the latter is likely altered by advected terrestrial nutrients from the glacier front. The glacial front samples
containing marine N in equal proportion to terrestrial N is consistent with entrainment and mixing that occurs during upwelling of the subglacial plume that draws nitrate from remnant winter cooled waters through lateral mixing as suggested by recent studies at Kronebreen and Kongsbreen fronts (Halbach et al., 2019; Hopwood et al., 2020).

Moreover, in Kongsfjorden, the contribution of terrestrial *vs* marine sources in the fjord basin to its nutrient inventory can be estimated using the average value of 4.4 $\pm$ 0.3 ‰-$\delta^{15}$N-NO$_3^-$ calculated from fjord basin samples below 100 m (**Fig.
6i**) avoiding isotopic fractionation during phytoplankton assimilation which is evident in shallower samples. In this case, marine contribution gains slight importance ($\frac{4.4-3.5}{((4.4\pm0.3)-3.5)+((5.1\pm0.1)-(4.4\pm0.3))} \times 100 = 56 \pm 3$ %) and terrestrial inputs remain significant (44 $\pm$ 3 %).



Errors in these estimates stem from applying annually integrated terrestrial endmember estimates from sediments to water column snapshot of marine and terrestrial mixing documented during this study. For example, terrestrial inputs from
early-season melt will give signatures closer to snowmelt, while late season incorporates heavier values due to denitrification in waterlogged soils and stronger washout of guano (Hayashi et al., 2018). Nevertheless, this assessment indicates that terrestrial sources of N make an important contribution to the dissolved pool of nitrate in the glacial front as well as the water column of the fjord.

In summary, we conclude that (i) snow and ice melt are the major sources of terrestrial nitrate to these fjords, (ii) at the glacial front, the nitrate is sourced from equal admixture of terrestrial and marine sources and (iii) terrestrial contribution remains significant to the whole fjord nutrient inventory even when considering deep waters. The prevalence of terrestrial N contribution at depth can be explained by N regeneration from sinking organic matter. Winter convection can also mix terrestrial N to the deepwater but this source is likely to be small. In the next section, nutrient regeneration is quantified relative
to winter convection which will allow understanding the relative contribution of terrestrial *vs* marine nutrients to overall fjord productivity.

**4.6 Terrestrial N contribution to primary productivity in Kongsfjorden during the summer**

Terrestrial inputs are an important contributor to the nutrient pool in the fjords, but what is its relative contribution to
productivity in the fjord? To estimate this we use the WCW in Kongsfjorden, which occupies the deep basin between the land and the sill, and has the longest residence time in Kongsfjorden as illustrated by high AOU and low oxygen saturation (**Fig. 4d, e**; Svendsen et al., 2002). Given low WCW renewal rates, this water mass should have received prolonged supply of regenerated nitrogen via settling particulate organic nitrogen with limited physical exchange with the surface and the isotopic signatures are not affected by biological uptake since water mass formation. We can therefore use WCW to quantify the
terrestrial contribution to productivity in the fjord as seen in the integrated regenerated nitrate pool and the preformed nitrate pool inherited by the water mass during formation through winter convection using dual isotopes of N and O.

This methodology exploits the difference in isotopic fractionation engendered during remineralisation and nitrification processes. The principle enabling this calculation is that, while the $\delta^{15}$N of remineralised $NO_3^-$ records isotopic signatures of
reactive N assimilated at the sea surface, $\delta^{18}$O of remineralised $NO_3^-$ is reset close to that of ambient water in which regeneration occurs. Using $\delta^{18}O_{WCW}$ (1.9 ± 0.2 ‰) measured in this study, ambient water $\delta^{18}O_{H2O}$=0.3‰ estimated in Kongsfjorden (MacLachlan et a., 2007) and $\delta^{18}O_{AW}$=2.8 ±0.3‰ previously measured for Atlantic waters outside Kongsfjorden (Tuerena et al., 2021b) we estimate 65 ± 15 % of the nitrate in the WCW is regenerated (see **Appendix C** for more details). Furthermore the estimated N isotopic compostion of this remineralised nitrate ($\delta^{15}N_{reg}$: 3.7- 4.1 ‰, **Appendix C**) demonstrates a major
contribution of terrestrial sources (63-88%) to fjord primary productivity and recycling as opposed to marine sources (12-37%) (summarised in **Table 1**). The higher proportion of terrestrial N in the regenerated N pool demonstrates its importance to primary production in the fjord and reflects the fact that terrestrial sources are supplied directly above the halocline throughout the summer growing season, whereas uptake of marine nutrients is hindered by the strong halocline that develops during the summer. This should be considered as a very broad estimate given the uncertainties associated with $\delta^{15}N_{reg}$ estimates. These
include (i) limited sampling points in WCW (n=3),(ii) empirical data used as an approximation of $\delta^{18}O_{AW}$ and $\delta^{18}O_{H2O}$ and (iii) the percentage of regenerated nitrate (65 ± 15%). Nevetheless, these estimates demonstrate the importance of terrestrial N inputs in supporting productivity (63-88%) while advection of marine nutrients is an important contributor (56 ± 3%) to the overall N inventory of Kongsfjorden (Hegseth et al., 2019). The former is consistent with McGovern et al. (2020) showing the





important influence of terrestrial nutrients on fjord productivity in Svalbard. The latter conclusion is supported by Hegseth et al. (2019).

One important aspect of this conclusion is that terrestrial N sources account for not only 63-88% of fjord productivity, but also for most of the regenerated pool of N in the deep isolated water masses of the fjord. These inputs are expected to increase with climate warming and with the relocation of N held in soils and permafrosts. Such enhanced terrestrial N inputs will have a

direct effect on surface productivity, N regeneration/cycling and the fjord N inventory. In extreme conditions this would result in oxygen depletion in the isolated water masses such as the WCW. An added effect is the retreat of tidewater glaciers and absence of vigorous circulation associated with subglacial plumes, which can further inhibit mixing, increase isolation of deep water masses and cause an expansion of oxygen depleted waters.

**4.7 Future changes**

Our results and analysis suggest terrestrial N inputs contribute to nearly one half of the summer nitrate inventory in these fjords and support a large proportion of the productivity in summer due to its input to the surface stratified layer. Secondly, the presence or absence of subglacial plume can influence the relative use of nitrate within the fjord and its exchange with the open ocean. Both of these aspects are subject to future change.


It is expected that future warming will increase the source and magnitude of terrestrial inputs with key implications to the nutrient inventory, fjord primary productivity and nutrient export offshore. Permafrost melting is likely to gain importance, mobilising nutrients and in turn increasing terrestrial input into the marine environment (Vonk et al, 2015). This study shows that terrestrial N contributions account for nearly half of the N inventory and much of the productivity in Kongsfjorden and

therefore, future increase in terrestrial N supply is likely to increase productivity as well as the storage and cycling of N within Arctic fjords.

This comparative study between two fjords illustates the importance of tidewater glaciers in nitrate use and cycling within the fjords. The widespread retreat of tidewater glaciers in the Arctic and Greenland (Østby et al., 2017; Slater et al., 2019) has

important implications on N exchange between fjords and the open ocean. It is expected that tidewater glaciers will retreat in Kongsfjorden to become land-terminating glaciers with strengthening Atlantification and warmer AW (Torsvik et al., 2019). The surface meltwater flow can decrease primary producitvity via light and nutrient limitation as documented in the case of Rijpfjorden. Ultimately, given that N is the limiting nutrient in the Arctic Ocean and marine settings in general, this would aid the export of N offshore and contribute to the overall net primary production in the Arctic Ocean (Terhaar et al., 2021).


The broad differences discussed here are likely to change with Arctic settings, for instance in Greenland fjords where terrestrial N supply is limited, due to the lack of N storage in soils, marine N supply is more important (Cape et al., 2019). Here subglacial plumes invigorate marine N supply through upwelling from depth in the fjord and this fuels much of the productivity (e.g., Meire et al., 2017). The retreat of glaciers can reduce this marine N supply from upwelling thus inhibiting productivity. In the

case of Svalbard fjords, terrestrial N supply is significant and is likely to gain further prominence in the future due to remobilisation of soil and permafrost N. In addition, we show that roughly half the N stored in deep fjord waters is also of terrestrial origin recycled within the fjord. Thus climate change is likely to increase productivity and N cycling and storage within the fjord. However, on longer time scales, if the distance between the fjord and the retreating glaciers becomes large enough, there will be more time for denitrification during transport, which may deplete the nutrient pool before it is discharged





into the fjord. Once glaciers fully melt, nutrient inputs will be mainly associated with seasonal snow melt with opportunities for enhanced N cycling and removal in soils potentially reducing nutrient inputs.

Regarding export of nutrients offshore, climate change can have contrasting results depending on the size of the fjords and its circulation. Increased terrestrial N inputs and freshwater discharge are expected to increase export rather than storage in
Rijpfjorden, as unused nutrients are flushed offshore given its shorter residence time. Thus, fjords with fast exchange rates like Rijpfjorden have more potential to alter pan-Arctic productivity as nutrients are exported offshore. In fjords like Kongsfjorden, additional N inputs in the future can increase terrestrial N recycled within the fjord as biological uptake here is more complete. In Kongsfjorden, recycled nutrients are stored in isolated waters such as the WCW and this pool will increase due to the larger nutrient supply. In addition, the retreat of tidewater glaciers has the effect of reducing mixing in the fjord and potential for
increasing retention of winter waters (Torsvik et al., 2019). The consequence of the combination of greater productivity fueled by terrestrial N, larger pool of N cycling within the fjord and reduced circulation in the absence of tidewater glaciers is reduced oxygen levels and possibly hypoxia in isolated deep waters of these fjords.

**Conclusion**

Kongsfjorden and Rijpfjorden are highly influenced by terrestrial nutrient inputs of N, P and Si. Particularly, terrestrial nitrate sources are key contributors to the $NO_3^-$ pool in the fjords and are roughly equal in proportion to marine contributions from AW. These terrestrial inputs carry excess silicate and phosphate relative to nitrate with respect to marine sources. Nitrate from snowpack and glacier melting are identified as dominant sources of terrestrial $NO_3^-$ in both fjords based on high $\delta^{18}O\text{-}NO_3^-$ and low $\delta^{15}N\text{-}NO_3^-$. In Kongsfjorden, biological uptake of these nutrients indicates that nitrate and possibly silicate limits
primary production despite terrestrial input. N limitation is partly caused by excess terrestrial P and Si limitation results from effective removal through the "silicate pump"; and stratification also contributes to both nutrient limitations. In Rijpfjorden, isotopic signatures indicate that nitrate is not fully utilised which may reflect light limitation and shallow stratification associated with the surface runoff. The contrast in nitrate use in these two fjords is attributed to the nature of meltwater inputs which is subglacial in Kongsfjorden and surface in Rijpfjorden and the resulting differences in circulation and stratification.
As a result Rijpfjorden contributes terrestrial $NO_3^-$ to the coastal sea whereas in Kongsfjorden terrestrial $NO_3^-$ is mainly retained and recycled within the fjord.

Given the significance of terrestrial nitrate sources in Svalbard fjords, it is postulated that continuing Arctic warming and enhanced meltwater discharge and terrestrial $NO_3^-$ inputs will impact fjordic primary productivity as well as the $NO_3^-$ exchange
offshore. The larger size and vigorous circulation associated with surface glacial plumes currently leads to complete use of nutrients in surface waters of Kongsfjorden. There, increase in terrestrial inputs and the retreat of tidewater glaciers due to climate change is postulated to increase productivity and nutrient cycling within the fjord possibly leading to more oxygen depletion in isolated deep waters which may expand given the less vigorous circulation in the absence the subglacial plume. In contrast, such changes in smaller fjords will lead to conditions akin to Rijpfjorden where surface inputs lead to strong
stratification and light limitation which limits $NO_3^-$ use and favours the export offshore of unused nutrients. This condition could become more common with ongoing glacial retreat in many Arctic fjords. The implication is that future increases in terrestrial nutrient inputs in smaller fjords such as Rijpfjorden has greater potential to impact primary productivity offshore by exporting the unused nutrients.






**Author contribution.** MSG and RSG wrote the paper. MSG measured nitrate isotopes for NP2017 and NP2018. RET and MCFD measured nitrate isotopes for JR17005 and FS2018, respectively. All authors contributed to the final version of the paper.

**Competing interests.** The authors declare that they have no conflict of interest.

**Acknowledgements.** This work resulted from the ARISE project (/P006310/1 awarded to RSG), part of the Changing Arctic Ocean programme, jointly funded by the UKRI Natural Environment Research Council (NERC) and the German Federal Ministry of Education and Research (BMBF). This study was also supported by the Norwegian Polar Institute, the Centre for
Ice, Climate, and Ecosystems (ICE) and the Research Council of Norway (project #244646). We would like to thank the captain and crew of the RV *Lance*.



**Table 1.** Summary of isotopic signatures (‰), endmember values (‰) and terrestrial and marine contributions (%) calculated in Kongsfjorden.

| REPORTED $\delta^{15}N$ SIGNATURES | ‰ | Terrestrial contribution (%) | Marine contribution (%) |
|---|---|---|---|
| Nitrate in upwelling plume (at glacier front) | 4.3 ± 0.1 | 50 ± 3 | 50 ± 3 |
| Nitrate in fjord basin (>100m depth) | 4.4 ± 0.3 | 44 ± 3 | 56 ± 3 |
| Regenerated nitrate (65 ± 15 %) in Winter Cooled Waters (integrated contribution) | 3.7 - 4.1 | 63-88 | 12-37 |
| Preformed nitrate (35 ± 15 %) in fjord basin - | Ranges betweeen 4.3 ±0.1 ‰ (mostly terrestrial) and 5.1 ±0.1 ‰ (fully marine) | | |

| ENDMEMBER VALUES | ‰ |
|---|---|
| Terrestrial $\delta^{15}N$ endmember. From *Kumar et al. (2018)* | 3.5 |
| Marine $\delta^{15}N$ endmember. From *Tuerena et al. (2021b)* | 5.1 ± 0.1 |
| WCW $\delta^{15}N$ value | 4.2 ± 0.2 |

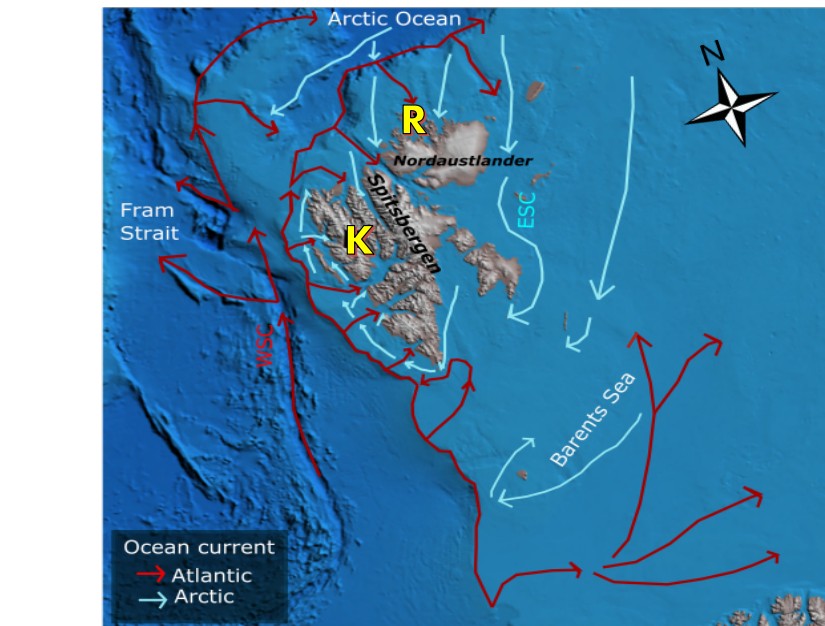

**Figure 1.** Norwegian archipelago of Svalbard (74-81ºN, 10-35ºE) with indication of Atlantic (red) and Arctic (blue) Water currents. K denotes Kongsfjorden; R denotes Rijpfjorden, WSC, West Spitsbergen Current; ESC, East Spitsbergen Current. The figure is based on Hop et al. (2019), Eriksen et al. (2018) and Leifer et al. (2018) and was designed using Global Mapper software (v.20.0, 2018), bathymetry data from IBCAO v.3 by Jakobsson et al. (2012).



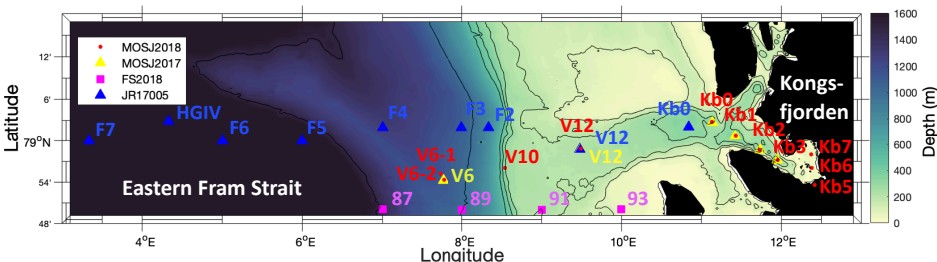

**Figure 2.** Bathymetric map of Kongsfjorden, Svalbard, and the eastern side of Fram Strait, illustrating sampling sites of the Norwegian Polar Institute Monitoring Cruises (2017, yellow triangles; 2018, red dots) for $\delta^{15}N$-$NO_3^-$ and $\delta^{18}O$-$NO_3^-$ determination. Samples were taken in the fjord (Kb0-3,5-7), as well as on the shelf (V12) and continental slope (V10 and V6). Data from sampling sites of FS2018 (pink squares) and JR17005 (blue triangles) cruises were also used for better coverage of offshore conditions. Shipboard measurements from cruise JR17005 were taken from the RSS *James Clark Ross* as part of the UK Changing Arctic Oceans programme in May-June 2018. Cruise FS2018 took place on the R.V *Kronprins Haakon* as part of the Fram Strait Arctic Outflow Observatory in August-September 2018.



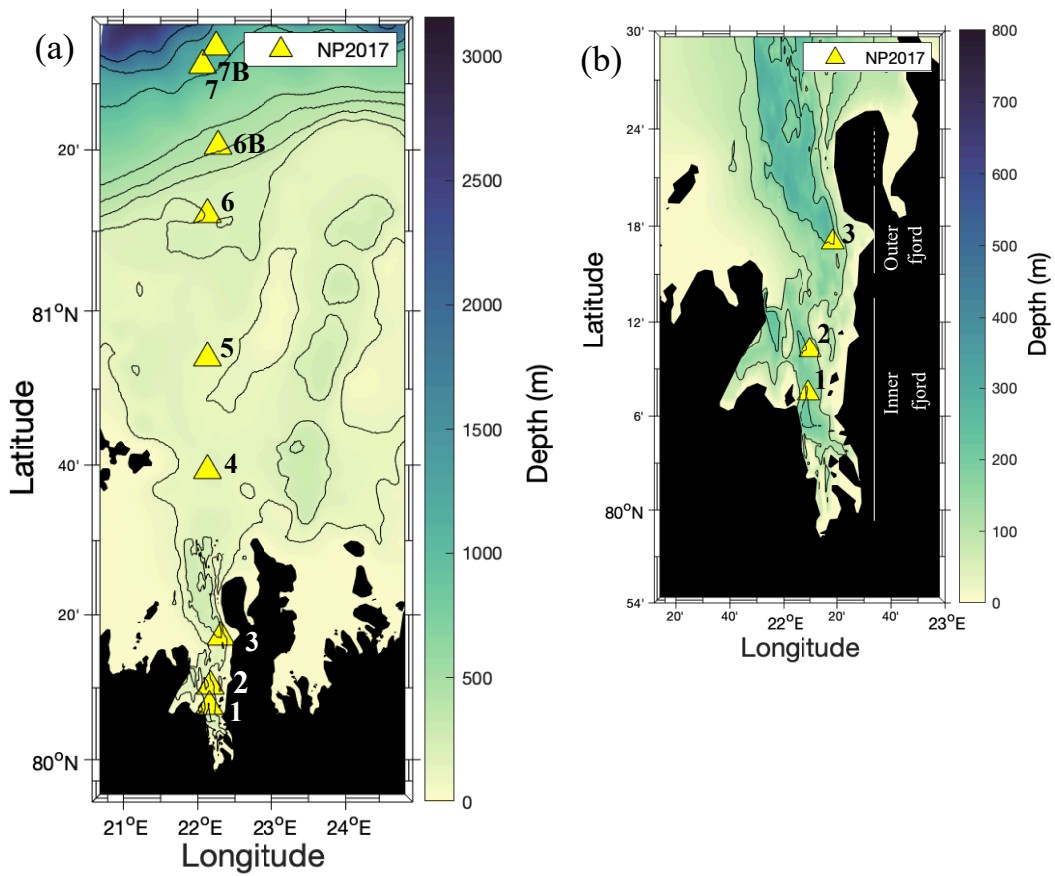

**Figure 3.** Bathymetric map of Rijpfjorden, Svalbard, illustrating sampling sites of the Norwegian Polar Institute monitoring cruises (NP2017, yellow triangles) for $\delta^{15}$N-NO$_3^-$ and $\delta^{18}$O-NO$_3^-$ determination. **(a)** All sample locations, namely in the inner fjord (R1-2), outer fjord (R3) as well as on the shelf (R4-R5) and continental slope (R6-R7B). **(b)** Zoom-in map of inner and outer fjord region.






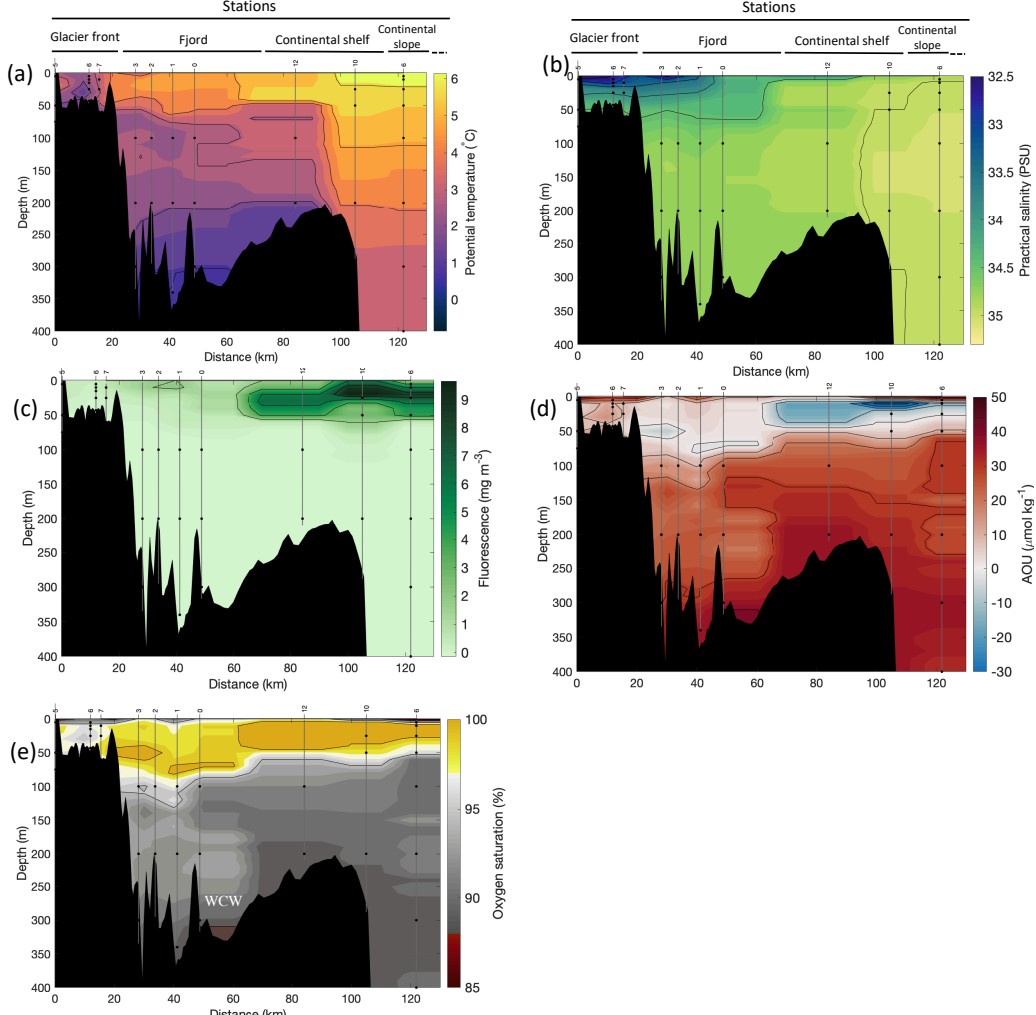

**Figure 4.** Cross-sections across Kongsfjorden of (**a**) potential temperature (ºC); (**b**) salinity (PSU); (**c**) fluorescence (mg m$^{-3}$), (**d**) apparent oxygen utilisation (AOU, µmol kg$^{-1}$) and (**e**) oxygen saturation (%). Data from CTD casts from NP2018 cruise. Grey vertical lines indicate the individual CTD casts where water samples were collected for isotopic analysis, with sampling
depths indicated by the black dots and station number indicated above each line. WCW; Winter Cooled Waters.



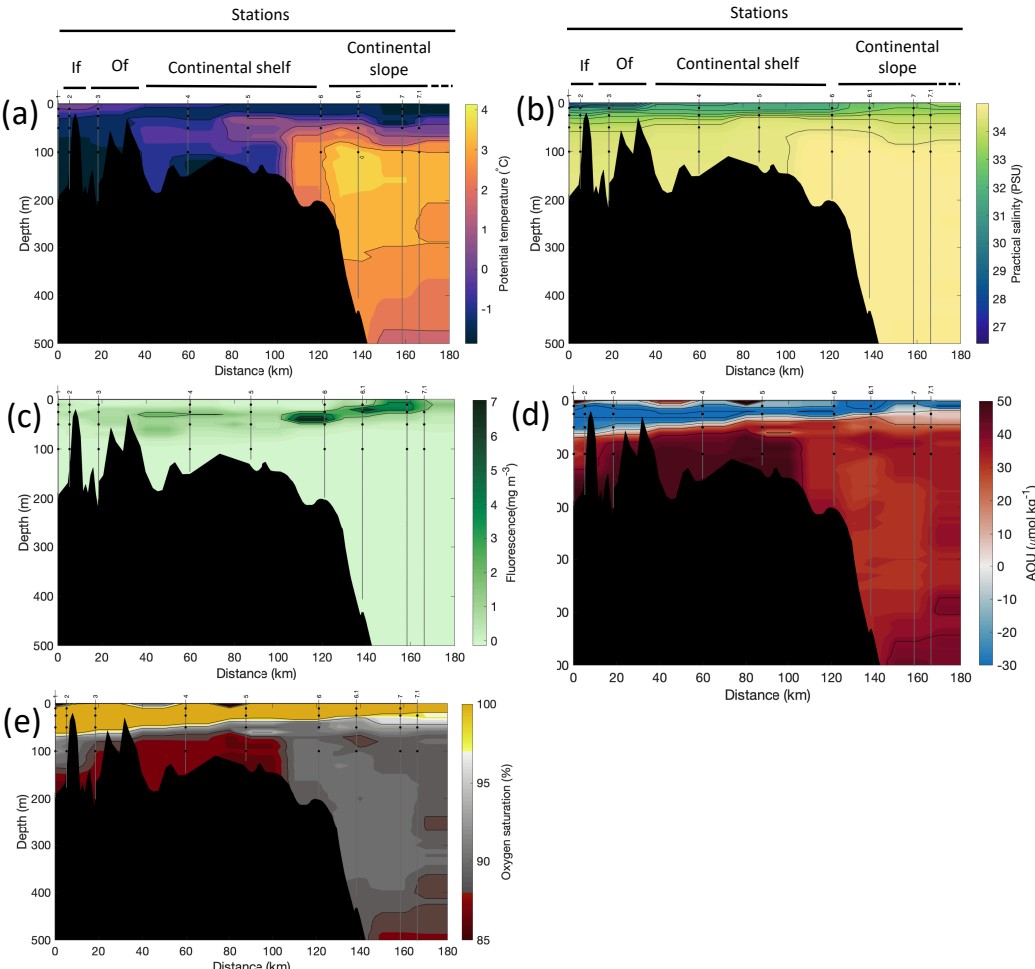

**Figure 5.** Cross-sections across Rijpfjorden of (**a**) potential temperature (ºC); (**b**) salinity (PSU); (**c**) fluorescence (mg m$^{-3}$), (**d**) apparent oxygen utilisation (AOU, μmol kg$^{-1}$), (**e**) oxygen saturation (%). Data from CTD casts from NP2017 cruise. Grey vertical lines indicate the individual CTD casts where water samples were collected for isotopic analysis, with sampling depths indicated by the black dots and station number indicated above each line. 'If', inner fjord; 'Of', outer fjord.





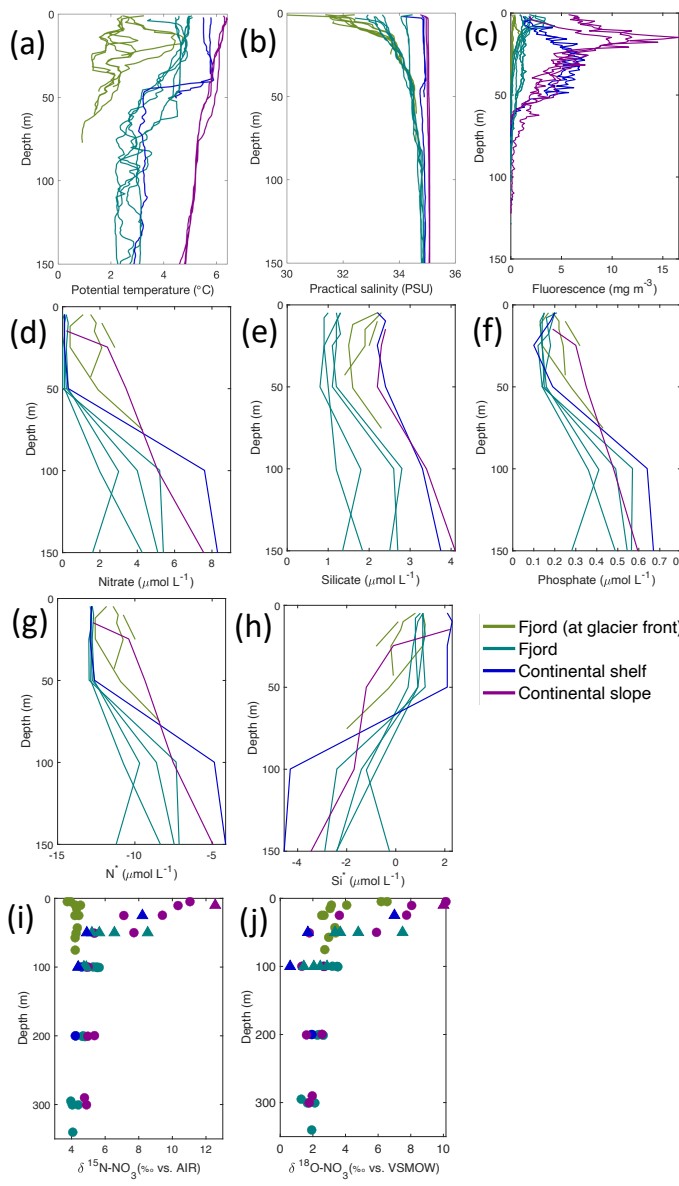


**Figure 6.** Depth profiles of (**a**) Temperature (ºC), (**b**) salinity (PSU), (**c**) fluorescence (mg L$^{-1}$), (**d**) nitrate (μmol L$^{-1}$), (**e**) silicate (μmol L$^{-1}$), (**f**) phosphate (μmol L$^{-1}$), (**g**) N*(μmol L$^{-1}$), (**h**) Si* (μmol L$^{-1}$), (**i**) $\delta^{15}$N-NO$_3^-$ (‰) and (**j**) $\delta^{18}$O-NO$_3^-$ (‰)
from Kongsfjorden using NP2018 (circles and solid lines) and NP2017 (triangles) cruise samples. Colour denotes locations in Kongsfjorden at glacier fronts, fjord, continental shelf and slope. Note: Standard deviations of the isotopic values reported in (**i**) and (**j**) are 0.33 and 0.47 ‰ for $\delta^{15}$N and $\delta^{18}$O, respectively.




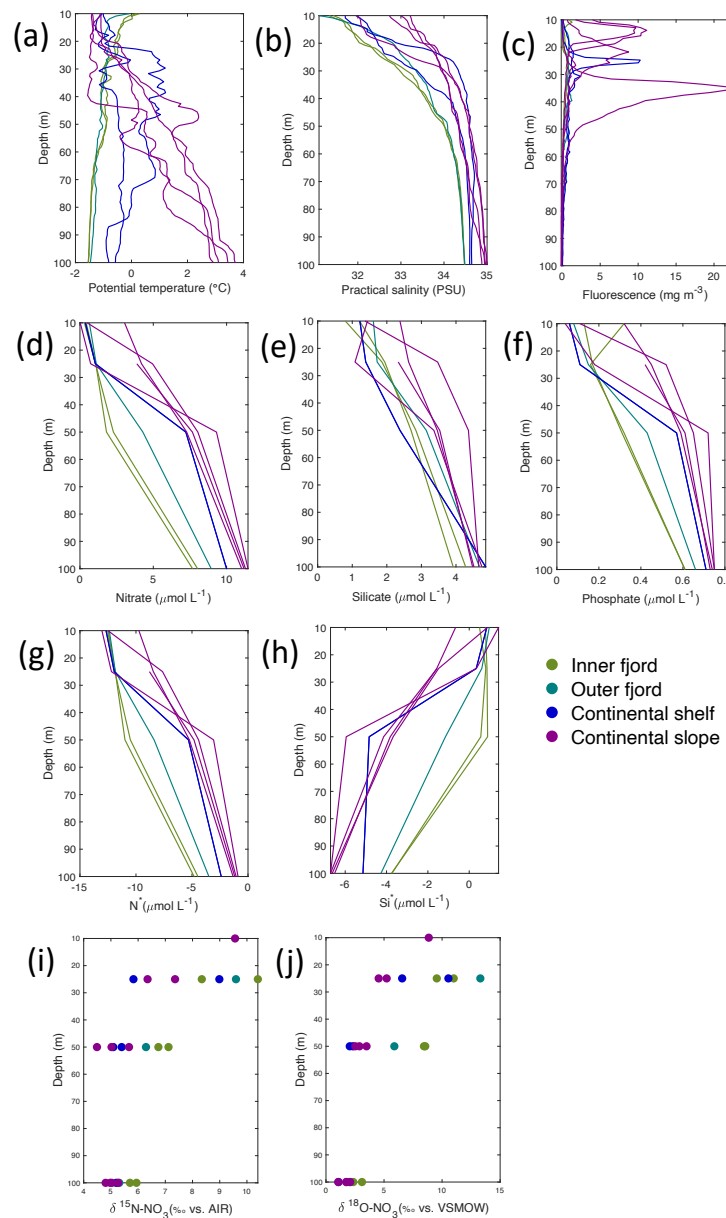

**Figure 7.** Depth profiles of (**a**) Temperature (ºC), (**b**) salinity (PSU), (**c**) fluorescence (mg m$^{-3}$), (**d**) nitrate (μmol L$^{-1}$), (**e**) silicate (μmol L$^{-1}$), (**f**) phosphate (μmol L$^{-1}$), (**g**) N*(μmol L$^{-1}$), (**h**) Si* (μmol L$^{-1}$), (**i**) $\delta^{15}$N-NO$_3^-$ (‰) and (**j**) $\delta^{18}$O-NO$_3^-$ (‰) from Rijpfjorden using NP2017 cruise samples. Colour denotes locations in Rijpfjorden (at inner fjord, outer fjord, continental shelf and slope). Note: Standard deviations of the isotopic values reported in (**i**) and (**j**) are 0.33 and 0.47 ‰ for $\delta^{15}$N and $\delta^{18}$O, respectively.



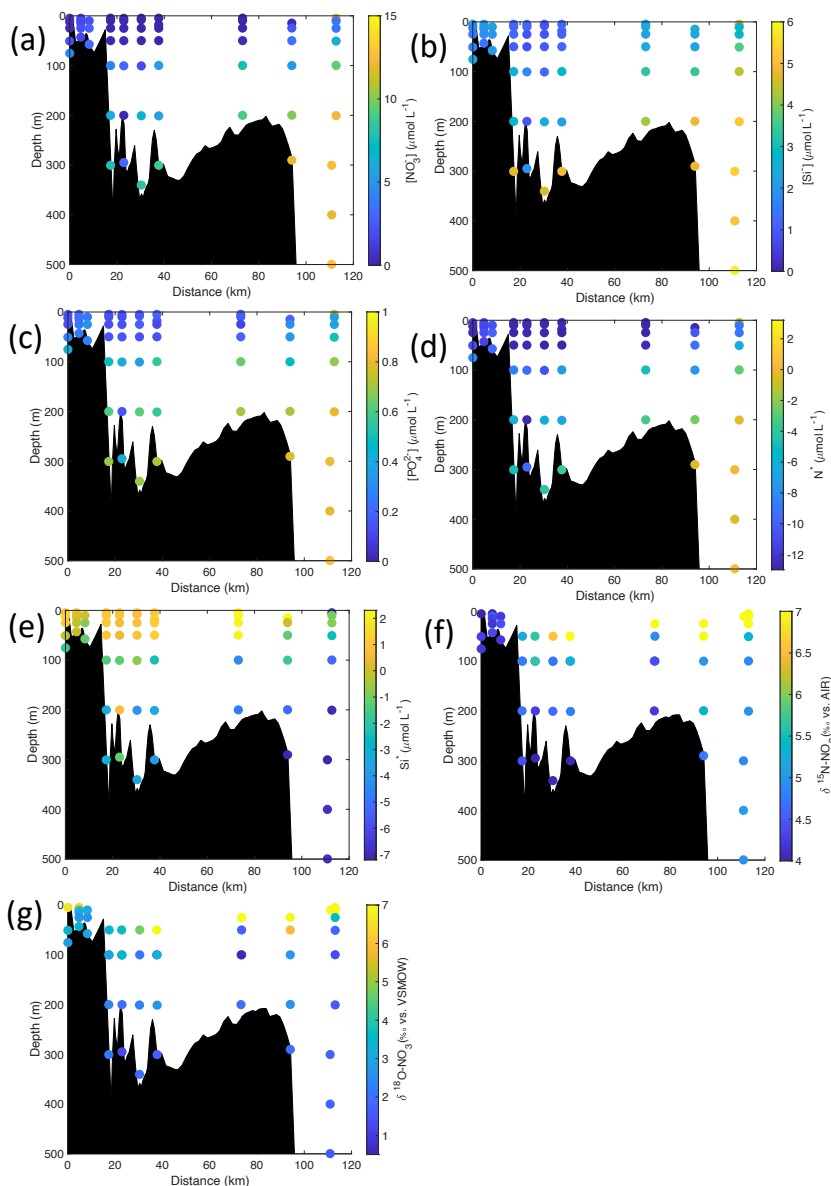

**Figure 8.** Cross-sections across Kongsfjorden of (**a**) nitrate (μmol L$^{-1}$), (**b**) silicate (μmol L$^{-1}$), (**c**) phosphate (μmol L$^{-1}$), (**d**) N* (μmol L$^{-1}$), (**e**) Si* (μmol L$^{-1}$), (**f**) $\delta^{15}$N-NO$_3^-$ (‰) and (**g**) $\delta^{18}$O-NO$_3^-$ (‰). Nutrient cross sections use data from NP2018 while isotope cross sections use NP2017 and NP2018 data. Note: Standard deviations of the isotopic values reported in (**f**) and (**g**) are 0.33 and 0.47 ‰ for $\delta^{15}$N and $\delta^{18}$O, respectively.






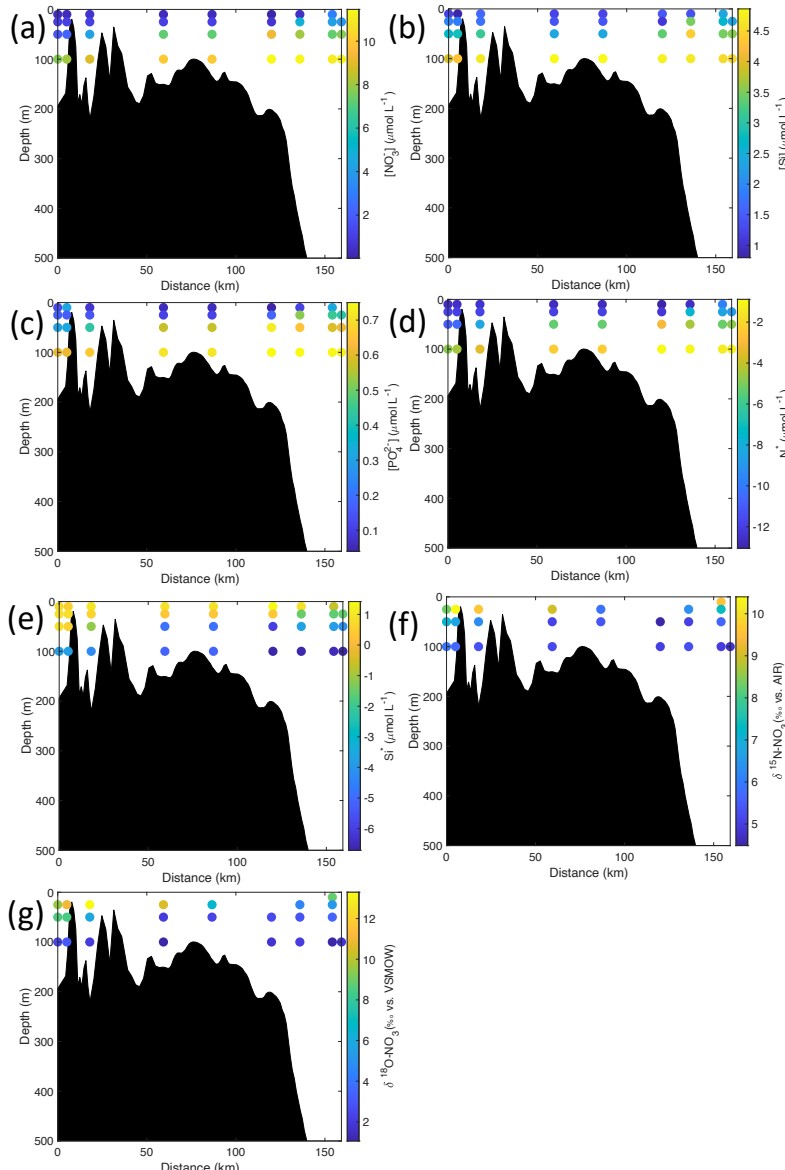

**Figure 9.** Cross-sections across Rijpfjorden of (**a**) nitrate (μmol L$^{-1}$), (**b**) silicate (μmol L$^{-1}$), (**c**) phosphate (μmol L$^{-1}$), (**d**)

N*(μmol L$^{-1}$), (**e**) Si* (μmol L$^{-1}$), (**f**) $\delta^{15}$N-NO$_3^-$ (‰) and (**g**) $\delta^{18}$O-NO$_3^-$ (‰) using NP2017 data. Note: Standard deviations of the isotopic values reported in (**f**) and (**g**) are 0.33 and 0.47 ‰ for $\delta^{15}$N and $\delta^{18}$O, respectively.





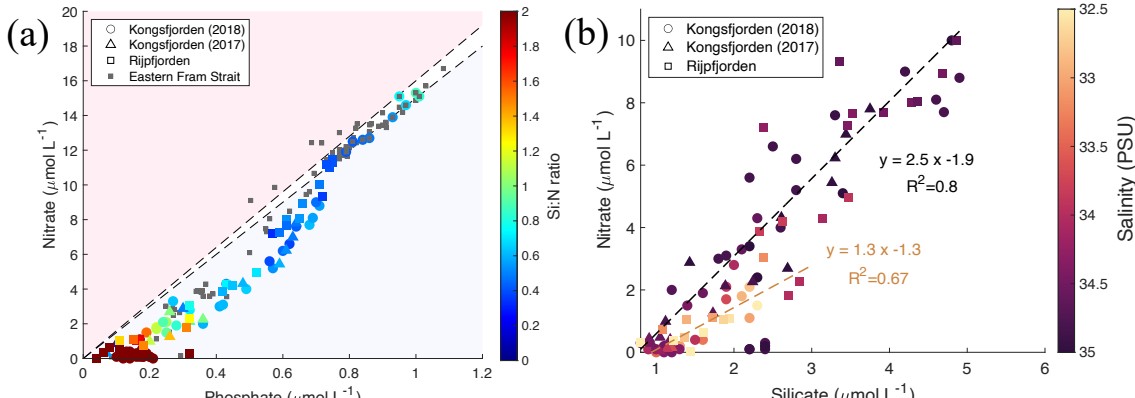

**Figure 10.** Correlation of nutrient concentrations throughout the whole water column of Kongsfjorden and Rijpfjorden. **(a)** nitrate (µmol L$^{-1}$) *vs.* phosphate (µmol L$^{-1}$) in relation to Redfield ratio (dashed lines for 1:15 and 1:16). In (a) colour coding for circles corresponds to Si:N ratio. Circles correspond to data measured in Kongsfjorden during NP2018 cruise. Triangles correspond to data measured in Kongsfjorden during NP2017 cruise. Squares correspond to data measured in Rijpfjorden during NP2017 cruise. Dark grey squares correspond to data from eastern Fram Strait (JR17005 and FS2018 cruises). **(b)** nitrate (µmol L$^{-1}$) *vs.* silicate (µmol L$^{-1}$) in Kongsfjorden (NP2018, circles; NP2017, triangles) and Rijpfjorden (NP2017, squares). Black regression line includes all points with salinity over 33.5 PSU. Light brown regression line includes all points with salinity below 33.5 PSU. In (b) colour coding corresponds to salinity. All *p*-values ≤ 0.05. Note: Regression results are in **Table B1**.

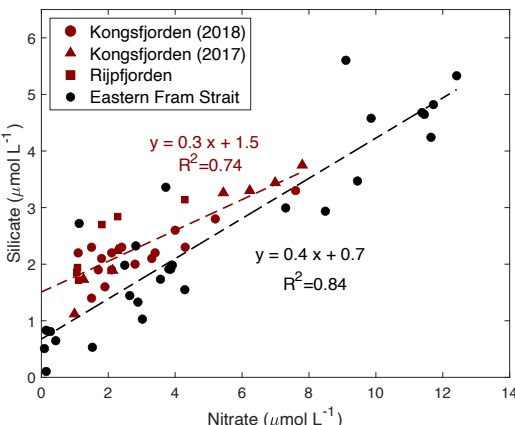

**Figure 11.** Correlation of silicate (µmol L$^{-1}$) *vs.* nitrate (µmol L$^{-1}$) in upper water column (<100 m) within the fjord (Kongsfjorden; NP2018 - dark red circles and NP2017- dark red triangles, Rijpfjorden; dark red squares) and outside of the fjord (eastern Fram Strait; black circles). Dark red dashed line is the regression line for Kongsfjorden (NP2018,2017) and Rijpfjorden (NP2017) surface data. Black dashed line is the regression line for eastern Fram Strait (JR17005 and FS2018) surface data. All *p*- values ≤0.05. Note: Regression results are in **Table B1**.





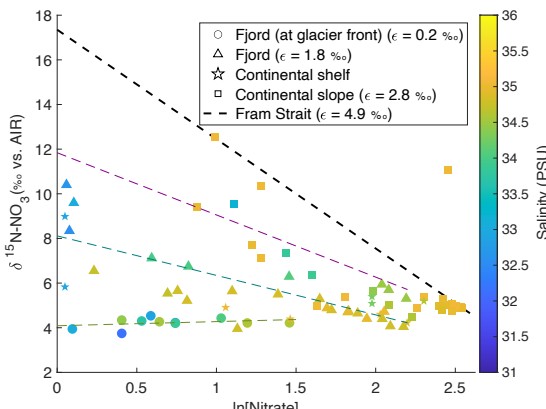


**Figure 12**. $\delta^{15}$N-NO$_3$ *vs.* ln (nitrate concentration) in the top 600 m in Kongsfjorden (NP2017, triangles; NP2018, circles) and Rijpfjorden (NP2017, squares) with salinity as a third variable. Shapes represent locations, namely at glacier fronts (circles), fjord (triangles), continental shelf (stars) and slope (squares). The gradients of the regression lines are representative of

fractionation factor ($\varepsilon$) for samples at glacier front (light green dotted line, $\varepsilon = 0.2$‰), fjord (dark green dotted line, $\varepsilon = 1.8$ ‰) and continental slope (magenta dotted line, $\varepsilon = 2.8$‰). Regression line for continental shelf samples was excluded due to insufficient data points. Black dotted line represents biological assimilation in Fram Strait where ε is equal to 4.9‰ as reported in Tuerena et al., (2021b). Equation of the line (y = -4.9x + 17.35) was estimated knowing that the gradient was equal to 4.9 and assuming initial $\delta^{15}$N-NO$_3^-$ of 5.1‰ and initial nitrate concentration of 11.8$\mu$M as reported in Tuerena et al (2021b). Note:

Standard deviations of the isotopic values reported are 0.33 and 0.47 ‰ for $\delta^{15}$N and $\delta^{18}$O, respectively.

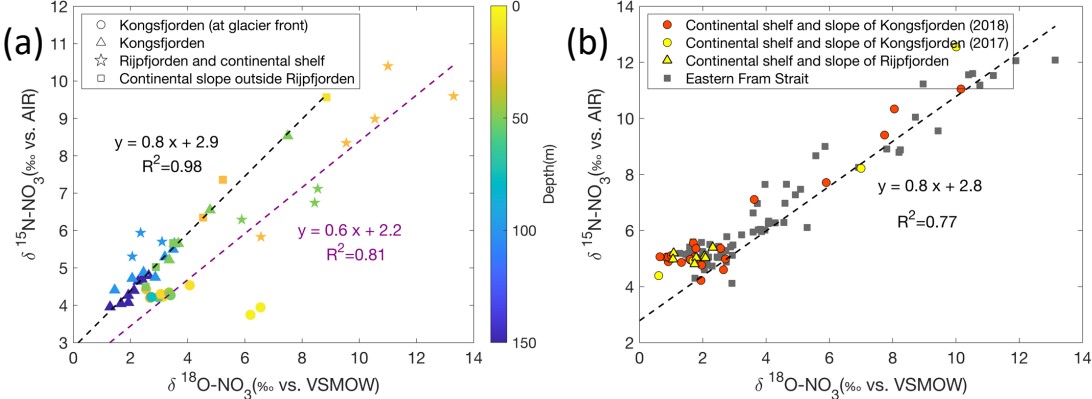


**Figure 13.** Correlation of $\delta^{15}$N-NO$_3$ *vs.* $\delta^{18}$O-NO$_3$ . **(a)** $\delta^{15}$N-NO$_3$ *vs.* $\delta^{18}$O-NO$_3$ and regression line (dashed line) for data reported in Kongsfjorden (NP2018, NP2017) and Rijpfjorden (NP2017). In (a) colour coding corresponds to depth and shapes represent different locations (at glacier fronts (O), Kongsfjorden (△), Rijpfjorden and its continental shelf (✶) and continental slope outside of Rijpfjorden (□)). Note that for the shelf and slope of Rijpfjorden, samples affected by AW intrusion were

excluded. Black dashed line is the regression line for all Kongsfjorden (NP2017-18) and continental slope stations outside of Rijpfjorden (not influenced by AW). Purple dashed line is the regression analysis for all Rijpfjorden and continental shelf surface values (not under the influence of AW). **(b)** $\delta^{15}$N-NO$_3$ *vs.* $\delta^{18}$O-NO$_3$, using data from all stations offshore of Kongsfjorden (continental shelf and slope- NP2018, red circles; NP2017, yellow circles), stations under AW influence offshore of Rijpfjorden (continental shelf and slope, yellow triangles) and from eastern Fram Strait (JR17005 and FS2018; dark grey



squares). Black dashed line is the regression line taken data from all offshore data (NP2018, NP2017, JR17005 and FS2018). All $p$-values ≤ 0.05. Note: Rijpfjorden samples under AW influence include shelf stations (i.e., R4, R5) below 25 m depth, and slope stations (i.e., R6, R6B and R7B) below 100 m.

Note: Regression results are in **Table B1**. Standard deviations of the isotopic values reported in **A** and **B** are 0.33 and 0.47 ‰ for $\delta^{15}N$ and $\delta^{18}O$, respectively.

**Appendix A- Supplementary Tables and Figures**

**Table A1.** Spatial distribution of the degree of stratification ($\Delta\rho$, kg m$^{-3}$) in Kongsfjorden and Rijpfjorden with an indication of location (at glacier fronts, fjord, continental shelf and slope) and stations.

| Kongsfjorden (MOSJ18) | Fjord (at glacier front) | | | Fjord | | | | Continental Shelf | Continental slope | | |
|---|---|---|---|---|---|---|---|---|---|---|---|
| | KB5 | KB6 | KB7 | KB3 | KB2 | KB1 | KB0 | V12 | V10 | V6 | |
| $\Delta\rho\left(\frac{kg}{m3}\right)$ | 2.7 ± 0.3 | | | 1.7 ± 0.5 | | | | 0.1 | 0.1 ± 0.1 | | |
| Rijpfjorden (MOSJ17) | | | | Fjord | | | Continental Shelf | | Continental slope | | | |
| | | | R1 | R2 | R3 | R4 | R5 | R6 | R6B | R7 | R7B |
| $\Delta\rho\left(\frac{kg}{m3}\right)$ | | | | 2.0 ± 0.2 | | 1.9 ± 0.3 | | | 1.2 ± 0.5 | | | |






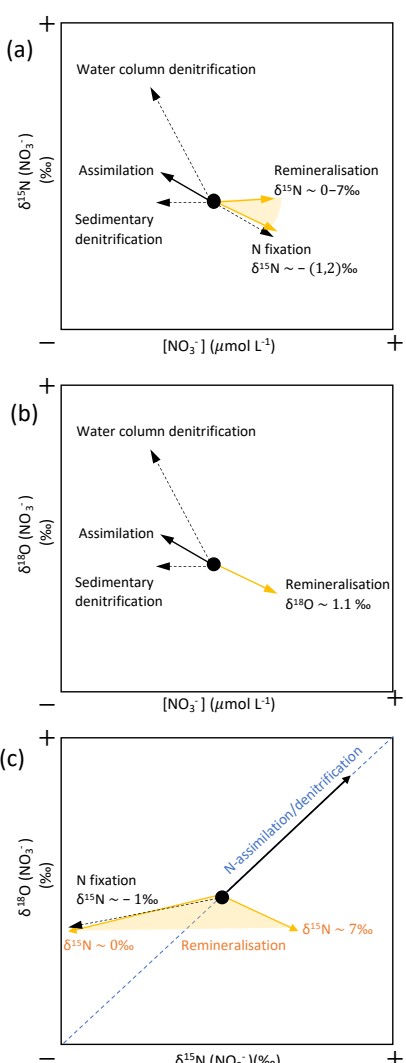

**Figure A1.** The effect of different marine N cycle processes on **(a)** $\delta^{15}$N-NO$_3^-$ and [NO$_3^-$], **(b)** $\delta^{18}$O-NO$_3^-$ and [NO$_3^-$] and **(c)** $\delta^{15}$N-NO$_3^-$ and $\delta^{18}$O-NO$_3^-$. The values of $\delta^{15}$N-NO$_3^-$ and $\delta^{18}$O-NO$_3^-$ designated by the black dot in (a), (b) and (c) represent global oceanic values, namely 5‰ and 2‰, respectively.

Dashed arrows denote processes that add or remove fixed N from the ocean, while solid arrows denote a component of the internal cycling of oceanic fixed N.

In (a), the orange shading represents variation of $\delta^{15}$N-NO$_3^-$ caused by the $\delta^{15}$N of the organic N being remineralised.

In (c), the blue dashed line represents the 1:1 ratio at which N-assimilation and denitrification occur. Moreover, the orange shading represents variation in the $\delta^{15}$N: $\delta^{18}$O ratio of the organic N being remineralised.

Compiled from Sigman and Fripiat (2019), Ryabenko (2013), Dähnke and Thamdrup (2013) and Sigman and Casciotti (2001).



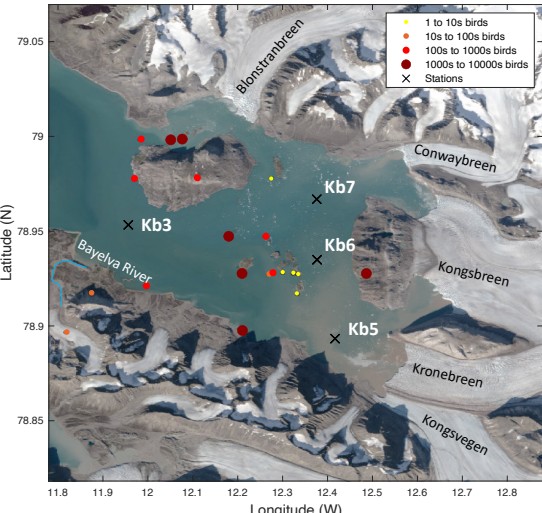

**Figure A2.** Seabird colonies, tidewater glaciers and river in Kongsfjorden, with an indication of the innermost NP2018 stations. Basemap Satellite image from the Norwegian Polar Institute (https://geodata.npolar.no/). Seabird colony data from Strøm et al (2008) and glacier names from Meslard et al. (2018).

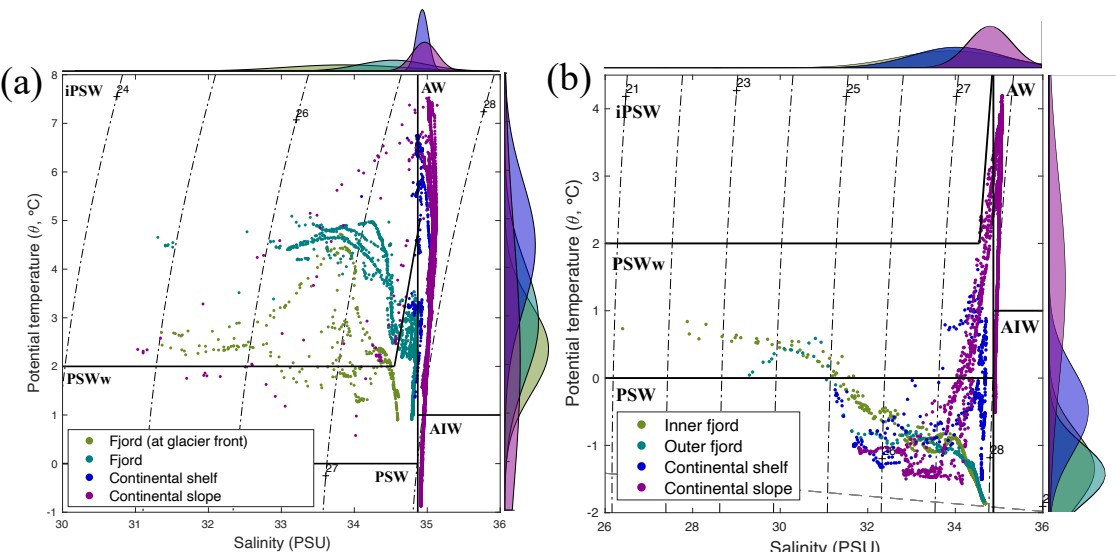

**Figure A3.** T-S diagrams for Kongsfjorden and Rijpfjorden study areas **(a)** Temperature-salinity plot for all CTD casts from NP2018 cruise taken in Kongsfjorden. Colours are indicative of different regions along Kongsfjorden, namely, light green illustrates the fjordic region at the glacier front (i.e., closest to land); dark green are the remaining stations in the fjord; dark blue is the shelf and magenta is the continental slope and offshore. Dashed lines represent density (kg m$^{-3}$). **(b)** Temperature-salinity plot for all CTD casts from NP2017 cruise taken in Rijpfjorden, Svalbard. Colours are indicative of different regions along Rijpfjorden, namely, light green illustrates the inner fjord region; dark green are the outer fjord stations; dark blue is the shelf and magenta is the continental slope and offshore.

Polygons indicate water types. AW, Atlantic Water; AIW, Arctic Intermediate Water; PSW, Polar Surface Water; iPSW, inshore Polar Surface Water; PSWw, warm Polar Surface Water. Marginal plots show density distribution of 1m binned





salinity (x-axis) and temperature (y-axis) values. Notice how at lower densities (shallower depths) polar surface waters are
dominant, whilst AW are more predominant at greater densities (greater depths). Water mass classification was based on that
by Pérez-Hernández et al. (2017).

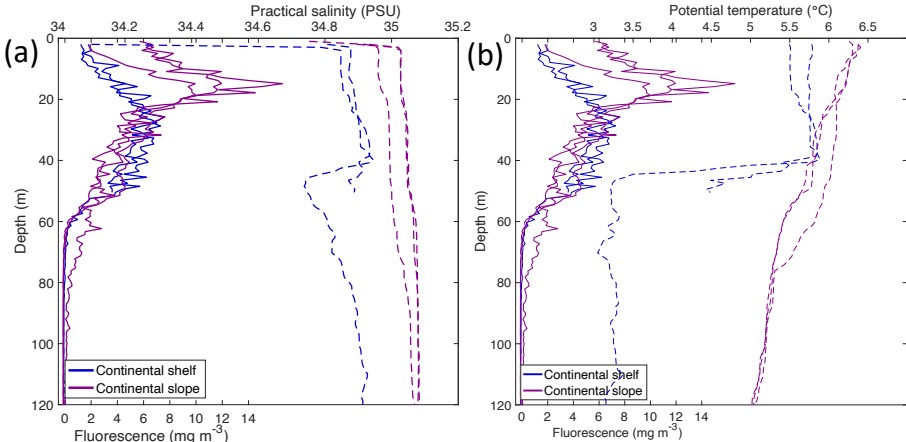

**Figure A4.** Profiles of **(a)** fluorescence (lines) and salinity (dashed lines) and **(b)** fluorescence (lines) and temperature (dashed lines). CTD data taken in continental shelf (blue) and slope (magenta) of Kongsfjorden during NP2018.


**Appendix B- Statistical analyses**

**Table B1**. Regression analyses (i.e., number of observations and *p*-values) for key linear models used in this study with an indication of the figure they correlate to.


| | Nº of observations | p-value |
|---|---|---|
| **Fig.10(a), Redfield ratio** | 127 | 1.82E-91 |
| **Fig.10(b), black line** | 57 | 5.61E-21 |
| **Fig.10(b), light brown line** | 22 | 3.07E-06 |
| **Fig.11, red line** | 30 | 9.57E-10 |
| **Fig.11, black line** | 29 | 3.43E-12 |
| **Fig.13(a), black line** | 25 | 4.95E-21 |
| **Fig.13(a), purple line** | 8 | 2.44E-03 |
| **Fig.13(b)** | 109 | 1.27E-35 |




**Appendix C- Calculations for the quantitative measure of the contribution of terrestrial inputs vs. marine inputs to the fjord nitrate pool**

**Step 1**- Calculating the percentage of regenerated nitrate in the Winter Cooled Water (WCW)

1)    Using equation in MacLachlan et al. (2007) (**Eqn.1**).

$$\boldsymbol{\delta^{18}O(H_2O)=0.43S-14.65}= 0.314 ‰ \qquad \textbf{Eqn.1}$$
where S in WCW= 34.8 ± 0.006 PSU (NP2018)

Previous field and modelling studies have used a nitrifying $\delta^{18}O$ value of **1.1‰ plus $\boldsymbol{\delta^{18}O_{H2O}}$** (Buchwald et al., 2012; Sigman et al., 2009; **Eqn.2**). Thus,

$$\delta^{18}O_{reg}(H_2O)=1.1‰ + 0.314‰ =1.4‰ \text{ in WCW} \qquad \textbf{Eqn.2}$$

2)    Using equation in Tuerena et al. (2021)

Since regeneration is calculated in the most saline and dense marine waters of the fjords the $\delta^{18}O$ values of Atlantic Waters can be used  (Granger et al., 2018; Tuerena et al., 2021). Thus, the following equation from Tuerena et al. (2021) (**Eqn.3**) was used to calculate the proportion of regenerated nitrate ($NO_3^-{}_{reg}/NO_3^-{}_{tot}$), and resulted in a value of 50-80% regeneration (65%

average).

$$\frac{NO3_{reg}}{NO3_{tot}} = \frac{(\delta^{18}O_{meas}-\delta^{18}O_{AW})}{(\delta^{18}O_{reg} - \delta^{18}O_{AW})} \qquad \textbf{Eqn.3}$$

$$= 65 \pm 15 \%$$


where $\delta^{18}O_{meas}= \delta^{18}O\text{-}NO_3^-$ measured in WCW= 1.9 ± 0.2‰ (data at >300 m, n=3) and $\delta^{18}O_{AW}=2.8 \pm 0.3‰$ (Tuerena et al., 2021). $\delta^{18}O_{reg}$ was 1.4‰, calculated using **Eqns.1-2** and $NO3_{reg}/NO3_{tot}=$ proportion of regenerated nitrate.

**Step 2** - Calculating $\delta^{15}N_{reg}$


The 65% regeneration estimate was then used to determine the $\delta^{15}N$ of remineralised $NO_3^-$ ($x$, **Eqn.4**). The $\delta^{15}N$ of preformed $NO_3^-$ is given as a range between the conservative estimate of 4.3 ± 0.1 ‰ (terrestrial estimate in this study) and 5.1 ± 0.1 ‰ (marine endmember; Tuerena et al., 2021) which assumes all preformed nitrate is of marine origin.

$$\delta^{15}N_{WCW} = \frac{65}{100}x + \left(\frac{35}{100} \times \delta^{15}N_{preformed}\right) \qquad \textbf{Eqn. 4}$$

where $x$ is $\delta^{15}N$ of remineralised $NO_3^-$,   $\delta^{15}N_{WCW} = \delta^{15}N\text{-}NO_3^-$ measured in WCW= 4.2 ± 0.2 ‰ (data at >300 m, n=3), $\delta^{15}N_{preformed}$ ranges between 4.3 ± 0.1 and  5.1 ± 0.1 ‰ (Tuerena et al., 2021), $\frac{65}{100}$ is the proportion of regenerated nitrate and $\frac{35}{100}$ is the proportion of preformed nitrate (derived from **Eqn. 5**).

**(1)  Conservative estimate (i.e., most of the preformed nitrate is terrestrial)**

Regenerated    Preformed

$$\delta^{15}N_{WCW =} \frac{65}{100}x + \left(\frac{35}{100} \times 4.3 \pm 0.1 ‰\right)$$

$$4.2\pm 0.2 ‰ = \frac{65}{100}x + \left(\frac{35}{100} \times 4.3 \pm 0.1 ‰\right)$$

$$x= \textbf{4.1} \pm \textbf{0.4 ‰}$$

**(2)  Assuming that all of the preformed nitrate is marine**

Regenerated    Preformed

$$\delta^{15}N_{WCW =} \frac{65}{100}x + \left(\frac{35}{100} \times 5.1 \pm 0.1 ‰\right)$$

$$4.2\pm 0.2 ‰ = \frac{65}{100}x + \left(\frac{35}{100} \times 5.1 \pm 0.1 ‰\right)$$

$$x= \textbf{3.7} \pm \textbf{0.4 ‰}$$





The resulting $\delta^{15}N$ values of 3.7- 4.1 ‰ of remineralised $NO_3^-$ demonstrates the major contribution of terrestrial sources (63-88%) to fjord primary productivity as opposed to marine sources (12-37%) (summarised in **Table 1**). This is explained by the fact that terrestrial sources occur in constant supply, whereas uptake of marine nutrients is hindered by the strong halocline that develops during the summer (**see section 4.6**).

**Literature cited in Appendix**

Buchwald, C., Santoro, A. E., Mcilvin, M. R. and Casciotti, K. L.: Oxygen isotopic composition of nitrate and nitrite produced by nitrifying cocultures and natural marine assemblages, , doi:10.4319/lo.2012.57.5.1361, 2012.

Dähnke, K. and Thamdrup, B.: Nitrogen isotope dynamics and fractionation during sedimentary denitrification in Boknis Eck, Baltic Sea, Biogeosciences, 10(5), 3079–3088, doi:10.5194/BG-10-3079-2013, 2013.

Granger, J., Sigman, D. M., Gagnon, J., Tremblay, J. E. and Mucci, A.: On the properties of the Arctic halocline and deep water masses of the Canada Basin from nitrate isotope ratios, J. Geophys. Res. Ocean., 123(8), 5443–5458, doi:10.1029/2018JC014110, 2018.

Maclachlan, S. E., Cottier, F. R., Austin, W. E. N. and Howe, J. A.: The salinity: δ18O water relationship in Kongsfjorden, western Spitsbergen, Polar Res., 26(2), 160–167, doi:10.1111/J.1751-8369.2007.00016.X, 2007.

Meslard, F., Bourrin, F., Many, G. and Kerhervé, P.: Suspended particle dynamics and fluxes in an Arctic fjord (Kongsfjorden, Svalbard), Estuar. Coast. Shelf Sci., 204, 212–224, doi:10.1016/J.ECSS.2018.02.020, 2018.

Pérez-Hernández, M. D., Pickart, R. S., Pavlov, V., Våge, K., Ingvaldsen, R., Sundfjord, A., Renner, A. H. H., Torres, D. J. and Erofeeva, S. Y.: The Atlantic Water boundary current north of Svalbard in late summer, J. Geophys. Res. Ocean., 122(3), 2269–2290, doi:10.1002/2016JC012486, 2017.

Ryabenko, E.: Stable Isotope Methods for the Study of the Nitrogen Cycle, Top. Oceanogr., doi:10.5772/56105, 2013.

Sigman, D. M. and Casciotti, K. L.: Nitrogen Isotopes in the Ocean, Encyclopedia of Ocean Sciences, pp. 1884–1894, Elsevier, doi:10.1006/rwos.2001.0172, 2001.

Sigman, D. M. and Fripiat, F.: Nitrogen isotopes in the ocean, Encycl. Ocean Sci., 263–278, doi:10.1016/B978-0-12-409548-9.11605-7, 2019.

Sigman, D. M., DiFiore, P. J., Hain, M. P., Deutsch, C. and Karl, D. M.: Sinking organic matter spreads the nitrogen isotope signal of pelagic denitrification in the North Pacific, Geophys. Res. Lett., 36(8), doi:10.1029/2008GL035784, 2009.

Strøm, H., Descamps, S., and Bakken, V.: Seabird Colonies by the Barents Sea, White Sea and Kara Sea, Norwegian Polar Institute, doi: 10.21334/npolar.2008.fd4fd3aa, 2008.

Tuerena, R. E., Hopkins, J., Ganeshram, R. S., Norman, L., De La Vega, C., Jeffreys, R. and Mahaffey, C.: Nitrate assimilation
and regeneration in the Barents Sea: Insights from nitrate isotopes, Biogeosciences, 18(2), 637–653, doi:10.5194/BG-18-637-2021, 2021.



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
