# Peer review of "Nitrate isotope investigations reveal future impacts of climate change on nitrogen inputs and cycling in Arctic fjords: Kongsfjorden and Rijpfjorden (Svalbard)"

_EGUsphere, 2022_

## Referee Comment (RC1)

**Comments on "Nitrate isotope investigations reveal future impacts of climate change on nitrogen inputs and cycling in Arctic fjords"**

By presenting hydrographic data, macro nutrients, and isotopic compositions of nitrate, Santos-Garcia et al. examined the influence of tidewater glaciers on nitrogen inputs and cycling in two fjords in Svalbard during the summertime. Overall, this study is a good demonstration of the application of nitrogen isotopes and nutrients in Arctic fjords to reveal the sources, cycling, and possible response of these nutrients to future changes in two well-characterized fjords. The work could be valuable to improve our understanding of the influence of climate change on polar region nutrient cycles. However, the structure of this manuscript needs to be further condensed and reorganized. The comments are as follows:

Major concerns:

**Major concerns:**

The **Introduction section** generally gives a good background to the study, but is structurally too long and does not flow smoothly from one section to the next (a subsection in the introduction is also awkward). It is recommended that the introductory section (e.g. Section 1.1 is too redundant) be streamlined and that the final section of the introduction briefly summarizes the shortcomings of current nutrient or nitrogen isotope studies in the Arctic fjords and surrounding waters to better introduce the purpose of the study.

**Section 3.2 titled Nutrient concentrations and isotopic ratios** actually contains descriptions of parameters such as temperature and salinity, while other parameters in Figures 4 and 5 (e.g. dissolved oxygen and chlorophyll) are not described. It is suggested that this could be split into two sections on environmental settings and nutrient/nitrogen isotope results respectively.

Throughout **the discussion section**, the language lacks conciseness, and the same idea or description appears repeatedly. For example, in lines 505-510, statements like "nitrate uptake is complete with near-zero nitrate values" appear several times, and similar phenomena occur repeatedly throughout the text. The narrative and language of

the article needed major improvement, and **the manuscript needed significant refinement of language and reduction in length.** Another suggestion is to consider merging some of the sub-section and renaming the subheadings. For example, the analysis in 4.1 and 4.2 is mainly both derived from the nutrient stoichiometry relationship, these two sections can be combined and the description of 4.2 needs to be simplified.

**Section 4.3**, where it is mentioned that the slope deviation from 1 in Figure 13 may be due to the regeneration of nitrate, i.e., the occurrence of nitrification, is it possible to assess the magnitude of this process? Does the fact that nitrification may introduce $^{15}N$ signatures from particulate nitrogen mineralization have an impact on the assessment of the source of nitrate? These need to be carefully evaluated.

In general, the article deals with two well-characterized fjords, involving relatively trivial descriptions of features. **Consider using a table or conceptual figure** to summarize the commonalities, differences, and responses of the two fjords in this paper in terms of hydrology and nutrient cycling under climate change.

Specific comments:

1. **Figures 1-3** are suggested to be merged into one figure, and Figure 1 should be labeled with latitude and longitude.

2. For the paragraphs, some indent the first line and some paragraphs don't, please keep the format consistent.

3. **Line 42:** "Nitrate is the predominant form of fixed N…" should be "Nitrate ($NO_3^-$) is the predominant form of fixed nitrogen (N)…"

4. **Line 70:** in Eqn.2 "$^{14}k/^{15}k$" should be "$^{14}k/^{15}k$"; **line 229** "1.8 ±0.2 (USGS 34)" should be "1.8±0.2 (USGS 34)"

5. **Line 455:** "The degree of stratification, and its associated nutrient dilution effect, can be ruled out as salinity alone could not explain the isotope effect in Figure 12." How was this conclusion reached? For example: if the isotope effect is due to the above factors, how should salinity be reflected in this graph?

6. **Figure 13a** Why is it important to represent depth by color, this message doesn't seem to be utilized much, at the moment this chart is a bit confusing, could different

colors be used to represent groups?

---

## Author Response (AR1)

**REPLY TO REFEREE #1**

We thank referee #1 for their constructive review of the manuscript which has helped improve the manuscript. Here we include responses to all of the comments as follows: (1) Reviewer's comment, (2) Author's comment and (3) Suggested changes to the manuscript.

(1). The Introduction section generally gives a good background to the study, but is structurally too long and does not flow smoothly from one section to the next (a subsection in the introduction is also awkward). It is recommended that the introductory section (e.g. Section 1.1 is too redundant) be streamlined and that the final section of the introduction briefly summarizes the shortcomings of current nutrient or nitrogen isotope studies in the Arctic fjords and surrounding waters to better introduce the purpose of the study.

(2). **Response**: We agree with the reviewer.

(3). **Action**: We have now streamlined and reduced the length of the introduction, reducing it to three subsections. In subsection 1 we outline the purpose of the study, we then provide a background of the study area in subsection 1.1 and an introduction to the isotopic methodology used is given in subsection 1.2 (as suggested by reviewer 2). In subsection 1.2, the equations that were previously used in the introduction -to explain the $\delta$ and $\varepsilon$ notation- are no longer included. These equations have now been moved to a new section in the Appendix ("Appendix B- Using stable isotope tools to determine N fluxes and cycling") as suggested by reviewer 2.

(1). Section 3.2 titled Nutrient concentrations and isotopic ratios actually contains descriptions of parameters such as temperature and salinity, while other parameters in Figures 4 and 5 (e.g. dissolved oxygen and chlorophyll) are not described. It is suggested that this could be split into two sections on environmental settings and nutrient/nitrogen isotope results respectively.

(2). **Response**: We agree with the reviewer.

(3). **Action**: We have included the environmental setting in the title of subsection 3.1. Discussion on relevant aspects of oxygen and chlorophyll distribution are in lines 238, 243 and 263-264. We retained the overall structure of the result section as it is in line with the succession of the figures introduced.

(1). Throughout the discussion section, the language lacks conciseness, and the same idea or description appears repeatedly. For example, in lines 505-510, statements like "nitrate uptake is complete with near-zero nitrate values" appear several times, and similar phenomena occur repeatedly throughout the text. The narrative and language of the article needed major improvement, and the manuscript needed significant refinement of language and reduction in length. Another suggestion is to consider merging some of the sub-section and renaming the subheadings. For example, the analysis in 4.1 and 4.2 is mainly both derived from the nutrient stoichiometry relationship, these two sections can be combined and the description of 4.2 needs to be simplified.

(2). **Response**: We agree with the reviewer.

(3). **Action**: We have attempted to rephrase and reduce redundancy wherever possible.

(1). Section 4.3, where it is mentioned that the slope deviation from 1 in Figure 13 may be due to the regeneration of nitrate, i.e., the occurrence of nitrification, is it possible to assess the magnitude of this process? Does the fact that nitrification may introduce 15N signatures from particulate nitrogen mineralization have an impact on the assessment of the source of nitrate? These need to be carefully evaluated.

(2). **Response**: Given that the slope of the line is so close to 1, regeneration is not the key process here and is not likely to have an important isotopic effect. Therefore, we have not tried to quantify this process. Similarly, in the glacial front, we cannot rule out nitrification. However, as shown in figure 13a, the d18O values here range from 2.3 to 6.5 per mil. These values are very heavy reflecting source

signatures rather than nitrification in the fjord which would move the values towards the water d18O of 0.3 per mil.
The clustering of d15N values at 4.3 per mil at the glacier front also suggest a source-dominated signature as nitrate uptake would make d15N heavier with decreasing depth.

(3). **Action**: No action taken.

(1). In general, the article deals with two well-characterized fjords, involving relatively trivial descriptions of features. Consider using a table or conceptual figure to summarize the commonalities, differences, and responses of the two fjords in this paper in terms of hydrology and nutrient cycling under climate change.

(2). **Response**. We thank the reviewer for this comment.

(3). **Action**: This has now been included as a new table 2 (referred to in line 699) as shown in the following page.

**Table 2.** Summary table of present conditions and future predictions of Kongsfjorden's and Rijpfjorden's hydrology and nutrient cycling characteristics.

| | Kongsfjorden | Rijpfjorden |
|---|---|---|
| | Sub-Arctic fjord (79.0°N, 11.7°E) | Arctic fjord (80.0°N, 22.3°E) |

**Present conditions**

**Physical characteristics**

| | Kongsfjorden | Rijpfjorden |
|---|---|---|
| Fjord size | ~26 km long, ~6-14 km wide | ~40 km long, ~7-12 km wide |
| Maximum fjord depth | Deeper (~350 m) | Shallower (~200 m) |
| Mean fjord depth | ~100 m | ~100 m |
| Continental shelf length | Narrower (~45 km) (∴ *stronger AW influence*) | Wider (~60 km) (∴ *weaker AW influence*) |

**Hydrology**

| | Kongsfjorden | Rijpfjorden |
|---|---|---|
| Glacial influence | Tidewater glaciers feed into fjord | Land-terminating glaciers feed into fjord |
| Riverine influence | Bayelva river | Small, short-lived rivers |
| Dominant type of freshwater discharge | Subsurface discharge | Surface discharge |
| Strength of fjordic circulation | Vigorous circulation (driven by subglacial plume) | Weaker vertical and lateral mixing |
| Estimated water residence time | ~172 hours (Yang et al., 2022) | Shorter (< 172 hours) |

**Nutrient cycling**

| | Kongsfjorden | Rijpfjorden |
|---|---|---|
| Stratification | Weaker - freshwater spread over a larger depth and is less fresh | Stronger -freshwater is confined to a fresher, thin layer extending further offshore |
| Light conditions | Weaker turbidity throughout fjord (except at glacier fronts) | Stronger turbidity throughout fjord |
| Nitrate utilisation | Complete | Partial |
| N storage and cycling | More N recycling within fjord→ Higher regenerated N storage | Less N recycling within fjord→ Lower regenerated N storage |
| Nitrate export offshore | Lower | Higher |

**Future predictions**

**Climate change effects**

| | Kongsfjorden | Rijpfjorden |
|---|---|---|
| Increased intrusion of warmer AW and higher air temperatures | → Encouraged tidewater glacier retreat | → Increased surface discharge |
| Increased permafrost/snowpack/glacier melting, increased riverine discharge and increased remobilisation of soils | → Increased terrestrial nitrate input to fjords (*note: in longer timescales, this might be reversed by increased denitrification rates*) | |

**Climate change consequences**

| | Kongsfjorden | Rijpfjorden |
|---|---|---|
| | *(while tidewater glaciers are still present)* | |
| Stratification | Increased | Increased |
| Fjord N inventory | Increased (fuelled primarily by terrestrial inputs) | Increased (fuelled primarily by terrestrial inputs) |
| Fjordic circulation | Reduced | Reduced |
| Light conditions | Increased turbidity | Increased turbidity |
| N storage and cycling | Increased | Decreased |
| Nitrate export offshore | Decreased | Increased |
| Oxygen levels at isolated deep waters | Hypoxia will potentially develop | |
| Net impact on primary productivity | Increased fjordic productivity | Increased productivity in offshore coastal regions |

**Specific comments:**

(1). **Figures 1-3** are suggested to be merged into one figure, and Figure 1 should be labeled with latitude

(2). **Res**

(3). **Act**
longitud

nd some paragraphs don't, please keep the format

d N…" should be "Nitrate (NO$_3^-$) is the predominant

(2). **Response**: We agree with the reviewer.

(3). **Action**: This has now been addressed (see line 42).

(1). **Line 70:** in Eqn.2 "$^{14}$k/ $^{15}$k" should be "$^{14}$k/$^{15}$k"; **line 229** "1.8 ±0.2 (USGS 34)" should be "1.8±0.2 (USGS 34)"

(2). **Response**: We agree with the reviewer.

(3). **Action**: This has now been addressed (see lines 959 and 199).

(1). **Line 455:** "The degree of stratification, and its associated nutrient dilution effect, can be ruled out as salinity alone could not explain the isotope effect in Figure 12." How was this conclusion reached? For example: if the isotope effect is due to the above factors, how should salinity be reflected in this graph?

(2). **Response**: Significant freshwater dilution reduces nitrate concentrations, thereby introducing an artifact in estimating the fractionation factor from the figure 12. If dilution has an important effect on nitrate concentrations, then you expect to see a linear relationship between salinity and nitrate concentration. However, this is not the case in this study (see new **Figure A4** below). Instead, nitrate changes are largely independent of salinity. Thus we can confirm that dilution effect has very little impact on nitrate concentrations and the resulting errors in fractionation factor deduced from d15N versus Ln Nitrate plot are limited (Fig. 12).

[Figure]

**Figure A4.** Nitrate concentrations vs salinity in the top 600m in Kongsfjorden (NP2017, triangles; NP2018, circles) and Rijpfjorden (NP2017, squares). Colours represent locations, namely at glacier fronts (light green) and fjord (dark green). Note how changes in nitrate concentrations are independent of salinity. Specifically, in (A) changes in salinity are not accompanied by a change in nitrate concentrations and vice versa in (B).

(3). **Action**: The figure above has now been included to justify our conclusion (referred to in line 425).

(1). **Figure 13a** Why is it important to represent depth by color, this message doesn't seem to be utilized much, at the moment this chart is a bit confusing, could different colors be used to represent groups?

(2). **Response**: in Figure 13a, the shapes of the symbols are already used to represent groups as indicated in the legend and the depth provides additional useful information since the system is stratified, biological uptake occurs in the upper layers and depth can be related to water masses

(3). **Action**: No action taken.

**REPLY TO REFEREE #2**

We thank referee #2 for their constructive review of the manuscript which has helped improve the manuscript. Here we include responses to all of the comments as follows: (1) Reviewer's comment, (2) Author's comment and (3) Suggested changes to the manuscript.

(1). While I feel all the visualizations (both in-text and supplementary plots) to be appealing, uncluttered, and informative, the opposite is true for the text presented in the manuscript. Most of the sentences and sections are unnecessarily cluttered and difficult to read. English usage and grammar need a major overhaul throughout the manuscript in my opinion.

(2). **Response**: We agree with the reviewer.

(3). **Action**: We have attempted to improve the text and its flow and have streamlined various parts of the text.

(1). Get rid of the equations in section 1.1. You may include them as a supplementary note. Reduce the number of sentences and also words within a sentence to better convey these ideas.

(2). **Response**: We agree with the reviewer.

(3). **Action**: We have now moved the equations to a new appendix (see Appendix B, lines 945-967).

(1). A1 is very good as it contains all the information/ideas (of section 1.1) one needs to follow the rest of the manuscript. The authors may move this inside the main text and move the equations to the supplementary file. This will enhance the reading experience.

(2). **Response**: We agree with the reviewer.

(3). **Action**: We have carried out this change in Section 1.2. This also now has a new figure 2.

(1). Why a particular water-mass classification scheme was adopted? Why not widely used Cottier et al., 2005?

(2). **Response**: The water-mass classification in Pérez-Hernández et al (2017) was adopted instead of that in Cottier et al (2005) because it provides definitions for the different varieties of PSW, and thus enabled a better idea of the proportions of the different freshwater inputs to the fjord. Recent studies have used the classification from Pérez-Hernández et al (2017) (e.g. Hop et al (2019)) for this reason and we have followed this to facilitate cross comparison. Nevertheless, the classification in Cottier et al (2005) was indeed used for a particular water mass, the Winter Cooled Waters, as this definition was essential for this study and it was not included in Pérez-Hernandez et al (2017).

(3). **Action**: No action taken.

(1). The x-axis of Fig 8 and several others (distance) is not clear. Please mark this distance as a line in Fig 2.

(2). **Response**: We agree with the reviewer.

(3). **Action**: We have carried out this change. Distance lines have now been included in new figure 1 (b)-(d) as shown below.

[Figure]

re 8 is cumulative distance (from one station to
dded to the figure caption.

ic. You may like to add the sampling period in
the text.

(5). **Action**: We have now included the sampling times 3- 5 August 2017 for Rijpfjorden and 13-15 July
for Kongsfjorden (see line 230). However, we decided not to include the sampling period throughout
the text to avoid cluttering.

(1). Lines (321-323): Why incomplete nutrient utilization? Wouldn't stratification enhance relative
nutrient utilization?

(2). **Response**: Indeed, stratification generally enhances relative nutrient utilisation, however, if there
is light limitation as we suggest to be the case in Rijpfjorden, this would retard nutrient utilisation. This
is mentioned in the discussion section 4.3 (lines 483-494, 507-525).

(3). **Action**: No action taken.

(1). Lines (349-353): What about low nutrient uptake close to glacier front and inner fjord in general.
Instead of solely attributing the cause to plume discharge, I would also see the prevailing axial
productivity gradient (see Kumar et al., 2016) as a potential reason. Perhaps it's a combination of both
these factors.

(2). **Response**: The axial productivity gradient in Kumar et al (2016) was explained by "nutrient laden
Atlantic water influx in the outer fjord region" leading to "better nutrient utilisation away from the glacier".
Our results do not necessarily contradict this as we show that 12-37 percent of recycled nitrate in the
deep waters is marine. However, during the study the nutrients in Kongsfjorden are completely depleted
(with the exception of glacial fronts) and thus we are unable to identify any gradients.

(3). **Action**: No action taken.

(1). Line 474: You may check a modified form of this equation, which uses more representative sampling (Tiwari et al., 2018, doi: 10.1016/j.gsf.2017.12.007).

(2). **Response**: We agree with the reviewer.

(3). **Action**: We agree that Tiwari et al (2018) uses more representative sampling. The equation in MacLachlan has now been replaced by that in Tiwari et al (2018), i.e., $\delta^{18}O_{H2O}$ = 0.54S-18.42, which also results in the $\delta^{18}O_{H2O}$ value of 0.3.

**REPLY TO PROF ANDY HODSON**

We thank Prof Andy Hodson for his comments which have helped improve and clarify the text of the manuscript. Here we include responses to all of the comments.

*The paper could consider using published seasonal (and multi-year) values of d15N-NO3 and d18O-NO3 in glacial runoff entering Kongsfjord. Linked to this, the paper could also be potentially improved by considering the role of nitrification (which is currently limited to a brief mention in the context of guano). Nitrification has been shown to become the dominant source of NO3 to glacial runoff in Kongsfjord after mid-July, and the "excess nitrate" it creates seems to be present in a worldwide selection of glaciers. While I am unsure of how this will affect the authors' important assertions about the future nitrogen balance of the two fjords being studied, I think it is really important to demonstrate a full appreciation of the role played by microorganisms in supplementing the nitrate content of runoff whilst glaciers retreat onto land. Two published studies of direct relevance to Kongsjorden are:*

*Wynn, P.M., Hodson, A.J., Heaton, T.H. and Chenery, S.R., 2007. Nitrate production beneath a High Arctic glacier, Svalbard. Chemical geology, 244(1-2), pp.88-102.*

*Ansari, A.H., Hodson, A.J., Kaiser, J. and Marca-Bell, A., 2013. Stable isotopic evidence for nitrification and denitrification in a High Arctic glacial ecosystem. Biogeochemistry, 113(1), pp.341-357.I*

*The above papers show that the inferred subglacial d15N-NO3 and d18O-NO3 end member signature in the discussion paper is quite different to those observed in glacial rivers. For example, during the main runoff season, subglacial d15N-NO3 was in the range -2 to -7 o/oo (Wynn et al). I am not entirely sure what this means for the discussion paper, but it would be good to see the authors' views on this and I hope it can help the discussion in Section 4.5, where I found sources mentioned that were difficult to understand (moulins?)*

**Response**: The emphasis of the paper is not on N cycling processes in the snowpack. Therefore, the extent to which we can discuss these aspects is limited. The light d15N values and heavy d18O values that we see in glacial front stations suggest a predominant source from atmospheric nitrate and cannot be explained by ammonia sourced N from snowpack. Also the terrestrial N endmember value that we use (based on Kumar et al., 2018) integrates over the season. Thus both this integrated value and the isotopes values documented during the study in both fjords suggests the predominance of this terrestrial nitrate sourced from ice melt. This is reflected in the manuscript with emphasis on this atmospheric nitrate source.

**Action**: However for completeness we now mention the papers suggested by the reviewer in the manuscript. We have now added a separate paragraph (lines 574-580) which mentions possible sources from ammonia and guano and the fact that these cannot be dominant sources since they should produce both lighter d15N and d18O values. In particular, the heavy d18O values (2.3 to 6.5 per mil) that we see in the glacial front can only come from atmospheric deposition on ice.

*Putting the strong seasonality of glacial outflow nitrate aside, I wonder whether the authors' inferred subglacial end-member requires more denitrification than is apparent from the published values of subglacial outflow. This might be because the dominant subglacial inputs to Kongsfjord come from far larger glaciers than those studied by Wynn et al and Ansari et al. I find this entirely plausible and also useful, because less denitrification after glacier retreat onto land is also a realistic proposition. It would also be good to question the representativeness of values from the smaller glaciers since they dominate the literature but not the inputs to fjords.*

**Response:** These are interesting points raised. Although we share the enthusiasm expressed by the reviewer we are unable to fully expand on what the reviewer suggests because (1) the paper is not about N cycling processes in ice and (2) due to constraint for space and comments by reviewers 1 and 2 on the length.

**Action:** We do mention seasonal variability and now have also included a comment on anoxia in the ice pack and referred to Wynn et al., 2007 (see lines 604-607).

Lastly, a minor point is that N2 fixation is indeed poorly understood, but it was studied on glaciers in the Kongsfjord region by the publication below. For sure, though, N2 fixation is not so important

Telling, J., Anesio, A.M., Tranter, M., Irvine-Fynn, T., Hodson, A., Butler, C. and Wadham, J., 2011. Nitrogen fixation on Arctic glaciers, Svalbard. *Journal of Geophysical Research: Biogeosciences*, *116*(G3).

**Action:** The reference Telling et al (2011) has been included when introducing the minor role of N2 fixation (see line 551).

---

## Referee Report (RR1)

**Comments on the 1st revision of Santos-Garcia et al. nitrate isotope paper**

I am pleased that the authors have greatly improved the quality of the manuscript in the last round of revisions and that the new tables summarize the theme of the article well. Overall, the current version with some minor changes, I think, can be accepted by the journal. But there is still some issues remaining need to be solved which are listed below.

The sub-sectioning in the 'introduction' is still a bit odd and this disrupts my reading. The latter two subsections could be considered to be integrated into the preceding paragraphs in the introduction, the 2.1 section, and the appendix. A concise introduction would allow the reader to move more quickly to the core of the results.

Lines 89-93 All these sentences are about the purpose of the article, please combine them. ("aims to…", "The purpose of…")

Line 137 "…possibly nutrient limitation in Rijpfjorden" More complete nutrient consumption (as you summarize in table 2) in Kongsfjorden may contradict this sentence?

Line 142 Casciotti et al., 2002 about oxygen isotope measurement should be cited.

> Casciotti, K.L., D.M. Sigman, M.G. Hastings, J.K. Böhlke, and A. Hilkert. 2002. "Measurement of the Oxygen Isotopic Composition of Nitrate in Seawater and Freshwater Using the Denitrifier Method". Analytical Chemistry 74: 4905-12. doi:10.1021/ac020113w.

Lines 187-188 "The denitrifier method for…" move this sentence into Line 194.

Lines 190-193 Move this paragraph after line 210.

Figures:

Fig. 1 There seems to be errors in the latitude and longitude of Fig. 1a, please confirm. Please double-check the latitude and longitude of the other figures as well. The y-axis of Fig. 1b is partially obscured

Fig. 3 Missing x-label in Fig. 3d (Distance (km)).

Fig. 7-8 Phosphate should be $PO_4^{3-}$ instead of $PO_4^{2-}$.

Fig. 11-12 A space should be added before the brackets (e.g., in x-labels and y-labels).

---

## Author Response (AR2)

We thank the anonymous reviewers for their comments in this second round of evaluation which have helped further improve the manuscript. Here we include responses to all comments as follows: (1) Reviewer's comment, (2) Author's comment and (3) Suggested changes to the manuscript.

(1). The sub-sectioning in the 'introduction' is still a bit odd and this disrupts my reading. The latter two subsections could be considered to be integrated into the preceding paragraphs in the introduction, the 2.1 section, and the appendix. A concise introduction would allow the reader to move more quickly to the core of the results.

(2). **Response**: We agree with the reviewer.

(3). **Action**: We have integrated subsection 1.1 into section 3.1 on environmental setting (lines 160- 211) and we have moved subsection 1.2 to the end of appendix C (lines 933- 955).

(1). Lines 89-93 All these sentences are about the purpose of the article, please combine them. ("aims to...", "The purpose of...")

(2). **Response**: We agree with the reviewer.

(3). **Action**: Done. See lines 88- 92.

(1). Line 137 "...possibly nutrient limitation in Rijpfjorden" More complete nutrient consumption (as you summarize in table 2) in Kongsfjorden may contradict this sentence?

(2). **Response**: We agree with the reviewer.

(3). **Action**: We have now re-phrased this sentence for clarification. See lines 209- 211.

(1). Line 142 Casciotti et al., 2002 about oxygen isotope measurement should be cited. Casciotti, K.L., D.M. Sigman, M.G. Hastings, J.K. Böhlke, and A. Hilkert. 2002. "Measurement of the Oxygen Isotopic Composition of Nitrate in Seawater and Freshwater Using the Denitrifier Method". Analytical Chemistry 74: 4905-12. doi:10.1021/ac020113w.

(2). **Response**: We agree with the reviewer.

(3). **Action**: Done. See line 934.

(1). Lines 187-188 "The denitrifier method for..." move this sentence into Line 194. Lines 190- 193 Move this paragraph after line 210.

(2). **Response**: We agree with the reviewer.

(3). **Action**: Done. See lines 119-120 and 138-140.

(1). Fig. 1 There seems to be errors in the latitude and longitude of Fig. 1a, please confirm.

Please double-check the latitude and longitude of the other figures as well. The y-axis of Fig. 1b is partially obscured

(2). **Response**: We agree with the reviewer. Thanks for the correction, Fig. 1a indeed showed errors in latitude and longitude caused by the polar projection. All other maps have Mercator projection and thus the latitude and longitude shown in figures are correct.

(3). **Action**: Done. See line 710.

(1). Fig. 3 Missing x-label in Fig. 3d (Distance (km)). Fig. 7-8 Phosphate should be PO43- instead of PO42-.

(2). **Response**: We agree with the reviewer.

(3). **Action**: Done. See lines 735, 790 and 805.

(1). Fig. 11-12 A space should be added before the brackets (e.g., in x-labels and y-labels).

(2). **Response**: We agree with the reviewer.

(3). **Action**: Done. See lines 835 and 850.

---

## Author Response (AR3)

Dear Editor,

The manuscript was revisited and no paragraphs consisting of only one or two sentences were found as said to be the case in lines 75, 190, 194, 206, 211-225, 230, etc. Perhaps the problem stems from re-sending the last manuscript version by One Drive (as requested by the file review validation team).

In any case, please find the final document enclosed.

Thank you very much.